# Frequency-domain MLPs are More Effective Learners in Time Series Forecasting

**Kun Yi**[1], **Qi Zhang**[2], **Wei Fan**[3], **Shoujin Wang**[4], **Pengyang Wang**[5], **Hui He**[1]
**Defu Lian**[6], **Ning An**[7], **Longbing Cao**[8], **Zhendong Niu**[1]*
[1]Beijing Institute of Technology, [2]Tongji University, [3]University of Oxford
[4]University of Technology Sydney, [5]University of Macau, [6]USTC
[7]HeFei University of Technology, [8]Macquarie University
{yikun, hehui617, zniu}@bit.edu.cn, zhangqi_cs@tongji.edu.cn, weifan.oxford@gmail.com
pywang@um.edu.mo, liandefu@ustc.edu.cn, ning.g.an@acm.org, longbing.cao@mq.edu.au

## Abstract

Time series forecasting has played the key role in different industrial, including finance, traffic, energy, and healthcare domains. While existing literatures have designed many sophisticated architectures based on RNNs, GNNs, or Transformers, another kind of approaches based on multi-layer perceptrons (MLPs) are proposed with simple structure, low complexity, and superior performance. However, most MLP-based forecasting methods suffer from the *point-wise mappings* and *information bottleneck*, which largely hinders the forecasting performance. To overcome this problem, we explore a novel direction of *applying MLPs in the frequency domain* for time series forecasting. We investigate the learned patterns of frequency-domain MLPs and discover their two inherent characteristic benefiting forecasting, (i) *global view*: frequency spectrum makes MLPs own a complete view for signals and learn global dependencies more easily, and (ii) *energy compaction*: frequency-domain MLPs concentrate on smaller key part of frequency components with compact signal energy. Then, we propose FreTS, a simple yet effective architecture built upon *Fre*quency-domain MLPs for *T*ime *S*eries forecasting. FreTS mainly involves two stages, (i) Domain Conversion, that transforms time-domain signals into *complex numbers* of frequency domain; (ii) Frequency Learning, that performs our redesigned MLPs for the learning of real and imaginary part of frequency components. The above stages operated on both inter-series and intra-series scales further contribute to channel-wise and time-wise dependency learning. Extensive experiments on 13 real-world benchmarks (including 7 benchmarks for short-term forecasting and 6 benchmarks for long-term forecasting) demonstrate our consistent superiority over state-of-the-art methods. Code is available at this repository: `https://github.com/aikunyi/FreTS`.

## 1   Introduction

Time series forecasting has been a critical role in a variety of real-world industries, such as climate condition estimation [1, 2], traffic state prediction [3, 4], economic analysis [5, 6], etc. In the early stage, many traditional statistical forecasting methods have been proposed, such as exponential smoothing [7] and auto-regressive moving averages (ARMA) [8]. Recently, the emerging development of deep learning has fostered many deep forecasting models, including Recurrent Neural Network-based methods (e.g., DeepAR [9], LSTNet [10]), Convolution Neural Network-based methods (e.g., TCN [11], SCINet [12]), Transformer-based methods (e.g., Informer [13], Autoformer [14]), and Graph Neural Network-based methods (e.g., MTGNN [15], StemGNN [16], AGCRN [17]), etc.

---

*Corresponding author

37th Conference on Neural Information Processing Systems (NeurIPS 2023).

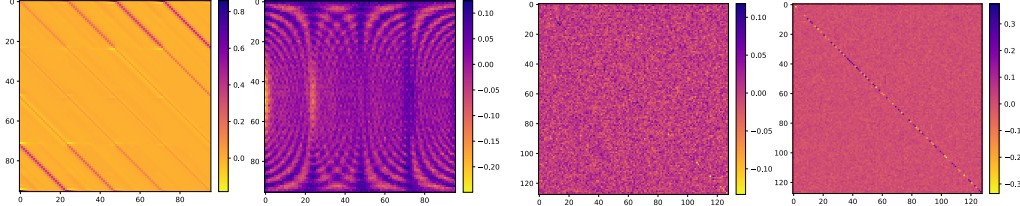

(a) **Left**: time domain. **Right**: frequency domain.    (b) **Left**: time domain. **Right**: frequency domain.

Figure 1: Visualizations of the learned patterns of MLPs in the time domain and the frequency domain (see Appendix B.4). (a) *global view*: the patterns learned in the frequency domain exhibits more obvious global periodic patterns than the time domain; (b) *energy compaction*: learning in the frequency domain can identify clearer diagonal dependencies and key patterns than the time domain.

While these deep models have achieved promising forecasting performance in certain scenarios, their sophisticated network architectures would usually bring up expensive computation burden in training or inference stage. Besides, the robustness of these models could be easily influenced with a large amount of parameters, especially when the available training data is limited [13, 18]. Therefore, the methods based on multi-layer perceptrons (MLPs) have been recently introduced with simple structure, low complexity, and superior forecasting performance, such as N-BEATS [19], LightTS [20], DLinear [21], etc. However, these MLP-based methods rely on *point-wise mappings* to capture temporal mappings, which cannot handle *global dependencies* of time series. Moreover, they would suffer from the *information bottleneck* with regard to the volatile and redundant *local momenta* of time series, which largely hinders their performance for time series forecasting.

To overcome the above problems, we explore a novel direction of *applying MLPs in the frequency domain* for time series forecasting. We investigate the learned patterns of frequency-domain MLPs in forecasting and have discovered their two key advantages: (i) *global view*: operating on spectral components acquired from series transformation, frequency-domain MLPs can capture a more complete *view* of signals, making it easier to learn *global* spatial/temporal dependencies. (ii) *energy compaction*: frequency-domain MLPs *concentrate* on the smaller key part of frequency components with the compact signal energy, and thus can facilitate preserving clearer patterns while filtering out influence of noises. Experimentally, we have observed that frequency-domain MLPs capture much more obvious global periodic patterns than the time-domain MLPs from Figure 1(a), which highlights their ability to recognize global signals. Also, from Figure 1(b), we easily note a much more clear diagonal dependency in the learned weights of frequency-domain MLPs, compared with the more scattered dependency learned by time-domain MLPs. This illustrates the great potential of frequency-domain MLPs to identify most important features and key patterns while handling complicated and noisy information.

To fully utilize these advantages, we propose FreTS, a simple yet effective architecture of *Fre*quency-domain MLPs for *T*ime *S*eries forecasting. The core idea of FreTS is to learn the time series forecasting mappings in the *frequency domain*. Specifically, FreTS mainly involves two stages: (i) Domain Conversion: the original time-domain series signals are first transformed into frequency-domain spectrum on top of Discrete Fourier Transform (DFT) [22], where the spectrum is composed of several *complex numbers* as frequency components, including the *real coefficients* and the *imaginary coefficients*. (ii) Frequency Learning: given the real/imaginary coefficients, we redesign the frequency-domain MLPs originally for the complex numbers by separately considering the real mappings and imaginary mappings. The respective real/imaginary parts of output learned by two distinct MLPs are then stacked in order to recover from frequency components to the final forecasting. Also, FreTS performs above two stages on both inter-series and intra-series scales, which further contributes to the channel-wise and time-wise dependencies in the frequency domain for better forecasting performance. We conduct extensive experiments on 13 benchmarks under different settings, covering 7 benchmarks for short-term forecasting and 6 benchmarks for long-term forecasting, which demonstrate our consistent superiority compared with state-of-the-art methods.

## 2  Related Work

**Forecasting in the Time Domain**    Traditionally, statistical methods have been proposed for forecasting in the time domain, including (ARMA) [8], VAR [23], and ARIMA [24]. Recently, deep learning

based methods have been widely used in time series forecasting due to their capability of extracting nonlinear and complex correlations [25, 26]. These methods have learned the dependencies in the time domain with RNNs (e.g., deepAR [9], LSTNet [10]) and CNNs (e.g., TCN [11], SCINet [12]). In addition, GNN-based models have been proposed with good forecasting performance because of their good abilities to model series-wise dependencies among variables in the time domain, such as TAMP-S2GCNets [4], AGCRN [17], MTGNN [15], and GraphWaveNet [27]. Besides, Transformer-based forecasting methods have been introduced due to their attention mechanisms for long-range dependency modeling ability in the time domain, such as Reformer [18] and Informer [13].

**Forecasting in the Frequency Domain**    Several recent time series forecasting methods have extracted knowledge of the frequency domain for forecasting [28]. Specifically, SFM [29] decomposes the hidden state of LSTM into frequencies by Discrete Fourier Transform (DFT). StemGNN [16] performs graph convolutions based on Graph Fourier Transform (GFT) and computes series correlations based on Discrete Fourier Transform. Autoformer [14] replaces self-attention by proposing the auto-correlation mechanism implemented with Fast Fourier Transforms (FFT). FEDformer [30] proposes a DFT-based frequency enhanced attention, which obtains the attentive weights by the spectrums of queries and keys, and calculates the weighted sum in the frequency domain. CoST [31] uses DFT to map the intermediate features to frequency domain to enables interactions in representation. FiLM [32] utilizes Fourier analysis to preserve historical information and remove noisy signals. Unlike these efforts that leverage frequency techniques to improve upon the original architecture such as Transformer and GNN, in this paper, we propose a new frequency learning architecture that learns both channel-wise and time-wise dependencies in the frequency domain.

**MLP-based Forecasting Models**    Several studies have explored the use of MLP-based networks in time series forecasting. N-BEATS [19] utilizes stacked MLP layers together with doubly residual learning to process the input data to iteratively forecast the future. DEPTS [33] applies Fourier transform to extract periods and MLPs for periodicity dependencies for univariate forecasting. LightTS [20] uses lightweight sampling-oriented MLP structures to reduce complexity and computation time while maintaining accuracy. N-HiTS [34] combines multi-rate input sampling and hierarchical interpolation with MLPs for univariate forecasting. LTSF-Linear [35] proposes a set of embarrassingly simple one-layer linear model to learn temporal relationships between input and output sequences. These studies demonstrate the effectiveness of MLP-based networks in time series forecasting tasks, and inspire the development of our frequency-domain MLPs in this paper.

## 3   FreTS

In this section, we elaborate on our proposed novel approach, FreTS, based on our redesigned MLPs in the frequency domain for time series forecasting. First, we present the detailed *frequency learning architecture* of FreTS in Section 3.1, which mainly includes two-fold frequency learners with domain conversions. Then, we detailedly introduce our redesigned *frequency-domain MLPs* adopted by above frequency learners in Section 3.2. Besides, we also theoretically analyze their superior nature of global view and energy compaction, as aforementioned in Section 1.

**Problem Definition**    Let $[X_1, X_2, \cdots, X_T] \in \mathbb{R}^{N \times T}$ stand for the regularly sampled multi-variate time series dataset with $N$ series and $T$ timestamps, where $X_t \in \mathbb{R}^N$ denotes the multi-variate values of $N$ distinct series at timestamp $t$. We consider a time series lookback window of length-$L$ at timestamp $t$ as the model input, namely $\mathbf{X}_t = [X_{t-L+1}, X_{t-L+2}, \cdots, X_t] \in \mathbb{R}^{N \times L}$; also, we consider a horizon window of length-$\tau$ at timestamp $t$ as the prediction target, denoted as $\mathbf{Y}_t = [X_{t+1}, X_{t+2}, \cdots, X_{t+\tau}] \in \mathbb{R}^{N \times \tau}$. Then the time series forecasting formulation is to use historical observations $\mathbf{X}_t$ to predict future values $\hat{\mathbf{Y}}_t$ and the typical forecasting model $f_\theta$ parameterized by $\theta$ is to produce forecasting results by $\hat{\mathbf{Y}}_t = f_\theta(\mathbf{X}_t)$.

### 3.1   Frequency Learning Architecture

The frequency learning architecture of FreTS is depicted in Figure 2, which mainly involves Domain Conversion/Inversion stages, Frequency-domain MLPs, and the corresponding two learners, i.e., the Frequency Channel Learner and the Frequency Temporal Learner. Besides, before taken to learners, we concretely apply a *dimension extension* block on model input to enhance the model capability. Specifically, the input lookback window $\mathbf{X}_t \in \mathbb{R}^{N \times L}$ is multiplied with a learnable

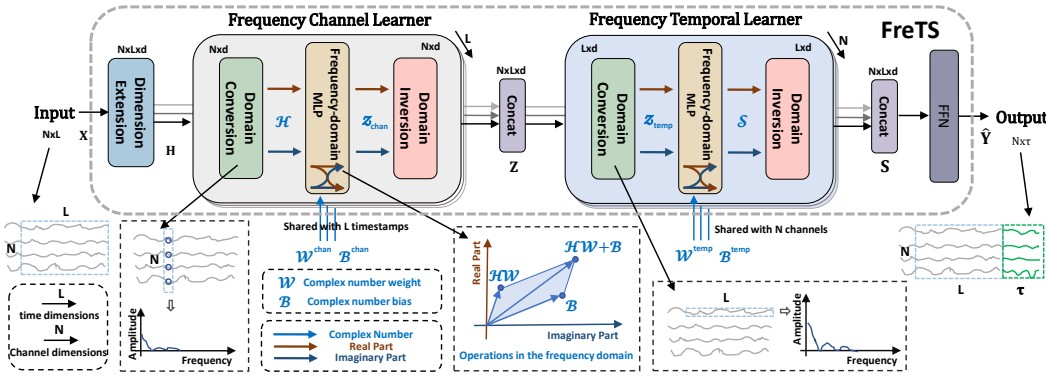

Figure 2: The framework overview of FreTS: the Frequency Channel Learner focuses on modeling inter-series dependencies with frequency-domain MLPs operating on the channel dimensions; the Frequency Temporal Learner is to capture the temporal dependencies by performing frequency-domain MLPs on the time dimensions.

weight vector $\phi_d \in \mathbb{R}^{1 \times d}$ to obtain a more expressive hidden representation $\mathbf{H}_t \in \mathbb{R}^{N \times L \times d}$, yielding $\mathbf{H}_t = \mathbf{X}_t \times \phi_d$ to bring more semantic information, inspired by word embeddings [36].

**Domain Conversion/Inversion**  The use of Fourier transform enables the decomposition of a time series signal into its constituent frequencies. This is particularly advantageous for time series analysis since it benefits to identify periodic or trend patterns in the data, which are often important in forecasting tasks. As aforementioned in Figure 1(a), learning in the frequency spectrum helps capture a greater number of periodic patterns. In view of this, we convert the input $\mathbf{H}$ into the frequency domain $\mathcal{H}$ by:

$$\mathcal{H}(f) = \int_{-\infty}^{\infty} \mathbf{H}(v)e^{-j2\pi fv}\mathrm{d}v = \int_{-\infty}^{\infty} \mathbf{H}(v)\cos(2\pi fv)\mathrm{d}v + j\int_{-\infty}^{\infty} \mathbf{H}(v)\sin(2\pi fv)\mathrm{d}v \quad (1)$$

where $f$ is the frequency variable, $v$ is the integral variable, and $j$ is the imaginary unit, which is defined as the square root of -1; $\int_{-\infty}^{\infty} \mathbf{H}(v)\cos(2\pi fv)\mathrm{d}v$ is the real part of $\mathcal{H}$ and is abbreviated as $Re(\mathcal{H})$; $\int_{-\infty}^{\infty} \mathbf{H}(v)\sin(2\pi fv)\mathrm{d}v$ is the imaginary part and is abbreviated as $Im(\mathcal{H})$. Then we can rewrite $\mathcal{H}$ in Equation (1) as: $\mathcal{H} = Re(\mathcal{H}) + jIm(\mathcal{H})$. Note that in FreTS we operate domain conversion on both the channel dimension and time dimension, respectively. Once completing the learning in the frequency domain, we can convert $\mathcal{H}$ back into the the time domain using the following inverse conversion formulation:

$$\mathbf{H}(v) = \int_{-\infty}^{\infty} \mathcal{H}(f)e^{j2\pi fv}\mathrm{d}f = \int_{-\infty}^{\infty} (Re(\mathcal{H}(f)) + jIm(\mathcal{H}(f))e^{j2\pi fv}\mathrm{d}f \quad (2)$$

where we take frequency $f$ as the integral variable. In fact, the frequency spectrum is expressed as a combination of cos and sin waves in $\mathcal{H}$ with different frequencies and amplitudes inferring different periodic properties in time series signals. Thus examining the frequency spectrum can better discern the prominent frequencies and periodic patterns in time series. In the following sections, we use DomainConversion to stand for Equation (1), and DomainInversion for Equation (2) for brevity.

**Frequency Channel Learner**  Considering channel dependencies for time series forecasting is important because it allows the model to capture interactions and correlations between different variables, leading to a more accurate predictions. The frequency channel learner enables communications between different channels; it operates on each timestamp by sharing the same weights between $L$ timestamps to learn channel dependencies. Concretely, the frequency channel learner takes $\mathbf{H}_t \in \mathbb{R}^{N \times L \times d}$ as input. Given the $l$-th timestamp $\mathbf{H}_t^{:,(l)} \in \mathbb{R}^{N \times d}$, we perform the *frequency channel learner* by:

$$\begin{aligned}
\mathcal{H}_{chan}^{:,(l)} &= \mathrm{DomainConversion}_{(chan)}(\mathbf{H}_t^{:,(l)}) \\
\mathcal{Z}_{chan}^{:,(l)} &= \mathrm{FreMLP}(\mathcal{H}_{chan}^{:,(l)}, \mathcal{W}^{chan}, \mathcal{B}^{chan}) \\
\mathbf{Z}^{:,(l)} &= \mathrm{DomainInversion}_{(chan)}(\mathcal{Z}_{chan}^{:,(l)})
\end{aligned} \quad (3)$$

where $\mathcal{H}_{chan}^{:,(l)} \in \mathbb{C}^{\frac{N}{2} \times d}$ is the frequency components of $\mathbf{H}_t^{:,(l)}$; $\text{DomainConversion}_{(chan)}$ and $\text{DomainInversion}_{(chan)}$ indicates such operations are performed along the channel dimension. FreMLP are frequency-domain MLPs proposed in Section 3.2, which takes $\mathcal{W}^{chan} = (\mathcal{W}_r^{chan} + j\mathcal{W}_i^{chan}) \in \mathbb{C}^{d \times d}$ as the complex number weight matrix with $\mathcal{W}_r^{chan} \in \mathbb{R}^{d \times d}$ and $\mathcal{W}_i^{chan} \in \mathbb{R}^{d \times d}$, and $\mathcal{B}^{chan} = (\mathcal{B}_r^{chan} + j\mathcal{B}_i^{chan}) \in \mathbb{C}^d$ as the biases with $\mathcal{B}_r^{chan} \in \mathbb{R}^d$ and $\mathcal{B}_i^{chan} \in \mathbb{R}^d$. And $\mathcal{Z}_{chan}^{:,(l)} \in \mathbb{C}^{\frac{N}{2} \times d}$ is the output of FreMLP, also in the frequency domain, which is conversed back to time domain as $\mathbf{Z}^{:,(l)} \in \mathbb{R}^{N \times d}$. Finally, we ensemble $\mathbf{Z}^{:,(l)}$ of $L$ timestamps into a whole and output $\mathbf{Z}_t \in \mathbb{R}^{N \times L \times d}$.

**Frequency Temporal Learner**   The frequency temporal learner aims to learn the temporal patterns in the frequency domain; also, it is constructed based on frequency-domain MLPs conducting on each channel and it shares the weights between $N$ channels. Specifically, it takes the frequency channel learner output $\mathbf{Z}_t \in \mathbb{R}^{N \times L \times d}$ as input and for the $n$-th channel $\mathbf{Z}_t^{(n),:} \in \mathbb{R}^{L \times d}$, we apply the *frequency temporal learner* by:

$$\mathcal{Z}_{temp}^{(n),:} = \text{DomainConversion}_{(temp)}(\mathbf{Z}_t^{(n),:})$$
$$\mathcal{S}_{temp}^{(n),:} = \text{FreMLP}(\mathcal{Z}_{temp}^{(n),:}, \mathcal{W}^{temp}, \mathcal{B}^{temp}) \tag{4}$$
$$\mathbf{S}^{(n),:} = \text{DomainInversion}_{(temp)}(\mathcal{S}_{temp}^{(n),:})$$

where $\mathcal{Z}_{temp}^{(n),:} \in \mathbb{C}^{\frac{L}{2} \times d}$ is the corresponding frequency spectrum of $\mathbf{Z}_t^{(n),:}$; $\text{DomainConversion}_{(temp)}$ and $\text{DomainInversion}_{(temp)}$ indicates the calculations are applied along the time dimension. $\mathcal{W}^{temp} = (\mathcal{W}_r^{temp} + j\mathcal{W}_i^{temp}) \in \mathbb{C}^{d \times d}$ is the complex number weight matrix with $\mathcal{W}_r^{temp} \in \mathbb{R}^{d \times d}$ and $\mathcal{W}_i^{temp} \in \mathbb{R}^{d \times d}$, and $\mathcal{B}^{temp} = (\mathcal{B}_r^{temp} + j\mathcal{B}_i^{temp}) \in \mathbb{C}^d$ are the complex number biases with $\mathcal{B}_r^{temp} \in \mathbb{R}^d$ and $\mathcal{B}_i^{temp} \in \mathbb{R}^d$. $\mathcal{S}_{temp}^{(n),:} \in \mathbb{C}^{\frac{L}{2} \times d}$ is the output of FreMLP and is converted back to the time domain as $\mathbf{S}^{(n),:} \in \mathbb{R}^{L \times d}$. Finally, we incorporate all channels and output $\mathbf{S}_t \in \mathbb{R}^{N \times L \times d}$.

**Projection**   Finally, we use the learned channel and temporal dependencies to make predictions for the future $\tau$ timestamps $\hat{\mathbf{Y}}_t \in \mathbb{R}^{N \times \tau}$ by a two-layer feed forward network (FFN) with one forward step which can avoid error accumulation, formulated as follows:

$$\hat{\mathbf{Y}}_t = \sigma(\mathbf{S}_t \phi_1 + \mathbf{b}_1)\phi_2 + \mathbf{b}_2 \tag{5}$$

where $\mathbf{S}_t \in \mathbb{R}^{N \times L \times d}$ is the output of the frequency temporal learner, $\sigma$ is the activation function, $\phi_1 \in \mathbb{R}^{(L*d) \times d_h}, \phi_2 \in \mathbb{R}^{d_h \times \tau}$ are the weights, $\mathbf{b}_1 \in \mathbb{R}^{d_h}$, $\mathbf{b}_2 \in \mathbb{R}^\tau$ are the biases, and $d_h$ is the inner-layer dimension size.

## 3.2   Frequency-domain MLPs

As shown in Figure 3, we elaborate our novel frequency-domain MLPs in FreTS that are redesigned for the complex numbers of frequency components, in order to effectively capture the time series key patterns with *global view* and *energy compaction*, as aforementioned in Section 1.

**Definition 1** (**Frequency-domain MLPs**). *Formally, for a complex number input $\mathcal{H} \in \mathbb{C}^{m \times d}$, given a complex number weight matrix $\mathcal{W} \in \mathbb{C}^{d \times d}$ and a complex number bias $\mathcal{B} \in \mathbb{C}^d$, then the frequency-domain MLPs can be formulated as:*

$$\mathcal{Y}^\ell = \sigma(\mathcal{Y}^{\ell-1}\mathcal{W}^\ell + \mathcal{B}^\ell)$$
$$\mathcal{Y}^0 = \mathcal{H} \tag{6}$$

*where $\mathcal{Y}^\ell \in \mathbb{C}^{m \times d}$ is the final output, $\ell$ denotes the $\ell$-th layer, and $\sigma$ is the activation function.*

As both $\mathcal{H}$ and $\mathcal{W}$ are complex numbers, according to the rule of multiplication of complex numbers (details can be seen in Appendix C), we further extend the Equation (6) to:

$$\mathcal{Y}^\ell = \sigma(Re(\mathcal{Y}^{\ell-1})\mathcal{W}_r^\ell - Im(\mathcal{Y}^{\ell-1})\mathcal{W}_i^\ell + \mathcal{B}_r^\ell) + j\sigma(Re(\mathcal{Y}^{\ell-1})\mathcal{W}_i^\ell + Im(\mathcal{Y}^{\ell-1})\mathcal{W}_r^\ell + \mathcal{B}_i^\ell) \tag{7}$$

where $\mathcal{W}^\ell = \mathcal{W}_r^\ell + j\mathcal{W}_i^\ell$ and $\mathcal{B}^\ell = \mathcal{B}_r^\ell + j\mathcal{B}_i^\ell$. According to the equation, we implement the MLPs in the frequency domain (abbreviated as FreMLP) by the separate computation of the real and imaginary parts of frequency components. Then, we stack them to form a complex number to acquire the final results. The specific implementation process is shown in Figure 3.

**Theorem 1.** *Suppose that **H** is the representation of raw time series and $\mathcal{H}$ is the corresponding frequency components of the spectrum, then the energy of a time series in the time domain is equal to the energy of its representation in the frequency domain. Formally, we can express this with above notations by:*

$$\int_{-\infty}^{\infty} |\mathbf{H}(v)|^2 \mathrm{d}v = \int_{-\infty}^{\infty} |\mathcal{H}(f)|^2 \mathrm{d}f \tag{8}$$

*where $\mathcal{H}(f) = \int_{-\infty}^{\infty} \mathbf{H}(v)e^{-j2\pi f v}\mathrm{d}v$, $v$ is the time/channel dimension, $f$ is the frequency dimension.*

We include the proof in Appendix D.1. The theorem implies that if most of the energy of a time series is concentrated in a small number of frequency components, then the time series can be accurately represented using only those components. Accordingly, discarding the others would not significantly affect the signal's energy. As shown in Figure 1(b), in the frequency domain, the energy concentrates on the smaller part of frequency components, thus learning in the frequency spectrum can facilitate preserving clearer patterns.

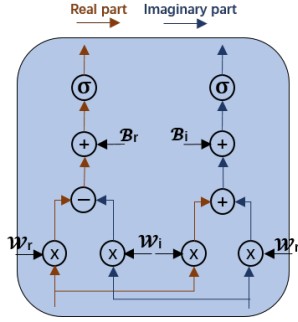

Figure 3: One layer of the frequency-domain MLPs.

**Theorem 2.** *Given the time series input **H** and its corresponding frequency domain conversion $\mathcal{H}$, the operations of frequency-domain MLP on $\mathcal{H}$ can be represented as global convolutions on **H** in the time domain. This can be given by:*

$$\mathcal{H}\mathcal{W} + \mathcal{B} = \mathcal{F}(\mathbf{H} * W + B) \tag{9}$$

*where $*$ is a circular convolution, $\mathcal{W}$ and $\mathcal{B}$ are the complex number weight and bias, $W$ and $B$ are the weight and bias in the time domain, and $\mathcal{F}$ is DFT.*

The proof is shown in Appendix D.2. Therefore, the operations of FreMLP, i.e., $\mathcal{H}\mathcal{W} + \mathcal{B}$, are equal to the operations $(\mathbf{H} * W + B)$ in the time domain. This implies that the operations of frequency-domain MLPs can be viewed as global convolutions in the time domain.

## 4  Experiments

To evaluate the performance of FreTS, we conduct extensive experiments on thirteen real-world time series benchmarks, covering short-term forecasting and long-term forecasting settings to compare with corresponding state-of-the-art methods.

**Datasets**  Our empirical results are performed on various domains of datasets, including traffic, energy, web, traffic, electrocardiogram, and healthcare, etc. Specifically, for the task of *short-term forecasting*, we adopt Solar [2], Wiki [37], Traffic [37], Electricity [3], ECG [16], METR-LA [38], and COVID-19 [4] datasets, following previous forecasting literature [16]. For the task of *long-term forecasting*, we adopt Weather [14], Exchange [10], Traffic [14], Electricity [14], and ETT datasets [13], following previous long time series forecasting works [13, 14, 30, 39]. We preprocess all datasets following [16, 13, 14] and normalize them with the min-max normalization. We split the datasets into training, validation, and test sets by the ratio of 7:2:1 except for the COVID-19 datasets with 6:2:2. More dataset details are in Appendix B.1.

**Baselines**  We compare our FreTS with the representative and state-of-the-art models for both short-term and long-term forecasting to evaluate their effectiveness. For short-term forecasting, we compre FreTS against VAR [23], SFM [29], LSTNet [10], TCN [11], GraphWaveNet [27], DeepGLO [37], StemGNN [16], MTGNN [15], and AGCRN [17] for comparison. We also include TAMP-S2GCNets [4], DCRNN [38] and STGCN [40], which require pre-defined graph structures, for comparison. For long-term forecasting, we include Informer [13], Autoformer [14], Reformer [18], FEDformer [30], LTSF-Linear [35], and the more recent PatchTST [39] for comparison. Additional details about the baselines can be found in Appendix B.2.

[2] https://www.nrel.gov/grid/solar-power-data.html
[3] https://archive.ics.uci.edu/ml/datasets/ElectricityLoadDiagrams20112014

**Implementation Details**   Our model is implemented with Pytorch 1.8 [41], and all experiments are conducted on a single NVIDIA RTX 3080 10GB GPU. We take MSE (Mean Squared Error) as the loss function and report MAE (Mean Absolute Errors) and RMSE (Root Mean Squared Errors) results as the evaluation metrics. For additional implementation details, please refer to Appendix B.3.

## 4.1   Main Results

Table 1: Short-term forecasting comparison. The best results are in **bold**, and the second best results are underlined. Full benchmarks of short-term forecasting are in Appendix F.1.

| Models | Solar | | Wiki | | Traffic | | ECG | | Electricity | | COVID-19 | |
|---|---|---|---|---|---|---|---|---|---|---|---|---|
| | MAE | RMSE | MAE | RMSE | MAE | RMSE | MAE | RMSE | MAE | RMSE | MAE | RMSE |
| VAR | 0.184 | 0.234 | 0.052 | 0.094 | 0.535 | 1.133 | 0.120 | 0.170 | 0.101 | 0.163 | 0.226 | 0.326 |
| SFM | 0.161 | 0.283 | 0.081 | 0.156 | 0.029 | 0.044 | 0.095 | 0.135 | 0.086 | 0.129 | 0.205 | 0.308 |
| LSTNet | 0.148 | 0.200 | 0.054 | 0.090 | 0.026 | 0.057 | 0.079 | 0.115 | 0.075 | 0.138 | 0.248 | 0.305 |
| TCN | 0.176 | 0.222 | 0.094 | 0.142 | 0.052 | 0.067 | 0.078 | 0.107 | 0.057 | 0.083 | 0.317 | 0.354 |
| DeepGLO | 0.178 | 0.400 | 0.110 | 0.113 | 0.025 | 0.037 | 0.110 | 0.163 | 0.090 | 0.131 | 0.169 | 0.253 |
| Reformer | 0.234 | 0.292 | 0.047 | 0.083 | 0.029 | 0.042 | 0.062 | 0.090 | 0.078 | 0.129 | 0.152 | 0.209 |
| Informer | 0.151 | 0.199 | 0.051 | 0.086 | 0.020 | 0.033 | 0.056 | 0.085 | 0.074 | 0.123 | 0.200 | 0.259 |
| Autoformer | 0.150 | 0.193 | 0.069 | 0.103 | 0.029 | 0.043 | 0.055 | 0.081 | 0.056 | 0.083 | 0.159 | 0.211 |
| FEDformer | 0.139 | 0.182 | 0.068 | 0.098 | 0.025 | 0.038 | 0.055 | 0.080 | 0.055 | 0.081 | 0.160 | 0.219 |
| GraphWaveNet | 0.183 | 0.238 | 0.061 | 0.105 | 0.013 | 0.034 | 0.093 | 0.142 | 0.094 | 0.140 | 0.201 | 0.255 |
| StemGNN | 0.176 | 0.222 | 0.190 | 0.255 | 0.080 | 0.135 | 0.100 | 0.130 | 0.070 | 0.101 | 0.421 | 0.508 |
| MTGNN | 0.151 | 0.207 | 0.101 | 0.140 | 0.013 | 0.030 | 0.090 | 0.139 | 0.077 | 0.113 | 0.394 | 0.488 |
| AGCRN | 0.123 | 0.214 | 0.044 | 0.079 | 0.084 | 0.166 | 0.055 | 0.080 | 0.074 | 0.116 | 0.254 | 0.309 |
| **FreTS (Ours)** | **0.120** | **0.162** | **0.041** | **0.074** | **0.011** | **0.023** | **0.053** | **0.078** | **0.050** | **0.076** | **0.123** | **0.167** |

Table 2: Long-term forecasting comparison. We set the lookback window size $L$ as 96 and the prediction length as $\tau \in \{96, 192, 336, 720\}$ except for traffic dataset whose prediction length is set as $\tau \in \{48, 96, 192, 336\}$. The best results are in **bold** and the second best are underlined. Full results of long-term forecasting are included in Appendix F.2.

| Models | FreTS | | PatchTST | | LTSF-Linear | | FEDformer | | Autoformer | | Informer | | Reformer | |
|---|---|---|---|---|---|---|---|---|---|---|---|---|---|---|
| Metrics | MAE | RMSE | MAE | RMSE | MAE | RMSE | MAE | RMSE | MAE | RMSE | MAE | RMSE | MAE | RMSE |
| **Weather** 96 | **0.032** | **0.071** | 0.034 | 0.074 | 0.040 | 0.081 | 0.050 | 0.088 | 0.064 | 0.104 | 0.101 | 0.139 | 0.108 | 0.152 |
| 192 | **0.040** | **0.081** | 0.042 | 0.084 | 0.048 | 0.089 | 0.051 | 0.092 | 0.061 | 0.103 | 0.097 | 0.134 | 0.147 | 0.201 |
| 336 | **0.046** | **0.090** | 0.049 | 0.094 | 0.056 | 0.098 | 0.057 | 0.100 | 0.059 | 0.101 | 0.115 | 0.155 | 0.154 | 0.203 |
| 720 | **0.055** | **0.099** | 0.056 | 0.102 | 0.065 | 0.106 | 0.064 | 0.109 | 0.065 | 0.110 | 0.132 | 0.175 | 0.173 | 0.228 |
| **Exchange** 96 | **0.037** | **0.051** | 0.039 | 0.052 | 0.038 | 0.052 | 0.050 | 0.067 | 0.050 | 0.066 | 0.066 | 0.084 | 0.126 | 0.146 |
| 192 | **0.050** | **0.067** | 0.055 | 0.074 | 0.053 | 0.069 | 0.064 | 0.082 | 0.063 | 0.083 | 0.068 | 0.088 | 0.147 | 0.169 |
| 336 | **0.062** | **0.082** | 0.071 | 0.093 | 0.064 | 0.085 | 0.080 | 0.105 | 0.075 | 0.101 | 0.093 | 0.127 | 0.157 | 0.189 |
| 720 | **0.088** | **0.110** | 0.132 | 0.166 | 0.092 | 0.116 | 0.151 | 0.183 | 0.150 | 0.181 | 0.117 | 0.170 | 0.166 | 0.201 |
| **Traffic** 48 | 0.018 | 0.036 | **0.016** | **0.032** | 0.020 | 0.039 | 0.022 | 0.036 | 0.026 | 0.042 | 0.023 | 0.039 | 0.035 | 0.053 |
| 96 | 0.020 | 0.038 | **0.018** | **0.035** | 0.020 | 0.042 | 0.023 | 0.044 | 0.033 | 0.050 | 0.030 | 0.047 | 0.035 | 0.054 |
| 192 | **0.019** | **0.038** | 0.020 | 0.039 | 0.020 | 0.040 | 0.022 | 0.042 | 0.035 | 0.053 | 0.034 | 0.053 | 0.035 | 0.054 |
| 336 | **0.020** | **0.039** | 0.021 | 0.040 | 0.021 | 0.041 | 0.021 | 0.040 | 0.032 | 0.050 | 0.035 | 0.054 | 0.035 | 0.055 |
| **Electricity** 96 | **0.039** | **0.065** | 0.041 | 0.067 | 0.045 | 0.075 | 0.049 | 0.072 | 0.051 | 0.075 | 0.094 | 0.124 | 0.095 | 0.125 |
| 192 | **0.040** | **0.064** | 0.042 | 0.066 | 0.043 | 0.070 | 0.049 | 0.072 | 0.072 | 0.099 | 0.105 | 0.138 | 0.121 | 0.152 |
| 336 | 0.046 | 0.072 | **0.043** | **0.067** | 0.044 | 0.071 | 0.051 | 0.075 | 0.084 | 0.115 | 0.112 | 0.144 | 0.122 | 0.152 |
| 720 | **0.052** | **0.079** | 0.055 | 0.081 | 0.054 | 0.080 | 0.055 | 0.077 | 0.088 | 0.119 | 0.116 | 0.148 | 0.120 | 0.151 |
| **ETTh1** 96 | **0.061** | **0.087** | 0.065 | 0.091 | 0.063 | 0.089 | 0.072 | 0.096 | 0.079 | 0.105 | 0.093 | 0.121 | 0.113 | 0.143 |
| 192 | **0.065** | **0.091** | 0.069 | 0.094 | 0.067 | 0.094 | 0.076 | 0.100 | 0.086 | 0.114 | 0.103 | 0.137 | 0.120 | 0.148 |
| 336 | 0.070 | **0.096** | 0.073 | 0.099 | **0.070** | 0.097 | 0.080 | 0.105 | 0.088 | 0.119 | 0.112 | 0.145 | 0.124 | 0.155 |
| 720 | **0.082** | **0.108** | 0.087 | 0.113 | **0.082** | **0.108** | 0.090 | 0.116 | 0.102 | 0.136 | 0.125 | 0.157 | 0.126 | 0.155 |
| **ETTm1** 96 | 0.052 | 0.077 | 0.055 | 0.082 | 0.055 | 0.080 | 0.063 | 0.087 | 0.081 | 0.109 | 0.070 | 0.096 | 0.065 | 0.089 |
| 192 | **0.057** | **0.083** | 0.059 | 0.085 | 0.060 | 0.087 | 0.068 | 0.093 | 0.083 | 0.112 | 0.082 | 0.107 | 0.081 | 0.108 |
| 336 | **0.062** | **0.089** | 0.064 | 0.091 | 0.065 | 0.093 | 0.075 | 0.102 | 0.091 | 0.125 | 0.090 | 0.119 | 0.100 | 0.128 |
| 720 | **0.069** | **0.096** | 0.070 | 0.097 | 0.072 | 0.099 | 0.081 | 0.108 | 0.093 | 0.126 | 0.115 | 0.149 | 0.132 | 0.163 |

**Short-Term Time Series Forecasting**   Table 1 presents the forecasting accuracy of our FreTS compared to thirteen baselines on six datasets, with an input length of 12 and a prediction length of 12. The best results are highlighted in bold and the second-best results are underlined. From the table, we observe that FreTS outperforms all baselines on MAE and RMSE across all datasets, and on average it makes improvement of 9.4% on MAE and 11.6% on RMSE. We credit this to the fact that FreTS explicitly models both channel and temporal dependencies, and it flexibly unifies channel and temporal modeling in the frequency domain, which can effectively capture the key patterns with the global view and energy compaction. We further report the complete benchmarks of short-term forecasting under different steps on different datasets (including METR-LA dataset) in Appendix F.1.

**Long-term Time Series Forecasting** Table 2 showcases the long-term forecasting results of FreTS compared to six representative baselines on six benchmarks with various prediction lengths. For the traffic dataset, we select 48 as the lookback window size $L$ with the prediction lengths $\tau \in \{48, 96, 192, 336\}$. For the other datasets, the input lookback window length is set to 96 and the prediction length is set to $\tau \in \{96, 192, 336, 720\}$. The results demonstrate that FreTS outperforms all baselines on all datasets. Quantitatively, compared with the best results of Transformer-based models, FreTS has an average decrease of more than 20% in MAE and RMSE. Compared with more recent LSTF-Linear [35] and the SOTA PatchTST [39], FreTS can still outperform them in general. In addition, we provide further comparison of FreTS and other baselines and report performance under different lookback window sizes in Appendix F.2. Combining Tables 1 and 2, we can conclude that FreTS achieves competitive performance in both short-term and long-term forecasting task.

## 4.2 Model Analysis

**Frequency Channel and Temporal Learners** We analyze the effects of frequency channel and temporal learners in Table 3 in both short-term and long-term experimental settings. We consider two variants: **FreCL**: we remove the frequency temporal learner from FreTS, and **FreTL**: we remove the frequency channel learner from FreTS. From the comparison, we observe that the frequency chan-

Table 3: Ablation studies of frequency channel and temporal learners in both short-term and long-term forecasting. 'I/O' indicates lookback window sizes/prediction lengths.

| Tasks | Short-term | | | | Long-term | | | |
|---|---|---|---|---|---|---|---|---|
| Dataset I/O | Electricity 12/12 | | METR-LA 12/12 | | Exchange 96/336 | | Weather 96/336 | |
| Metrics | MAE | RMSE | MAE | RMSE | MAE | RMSE | MAE | RMSE |
| FreCL | 0.054 | 0.080 | 0.086 | 0.168 | 0.067 | 0.086 | 0.051 | 0.094 |
| FreTL | 0.058 | 0.086 | 0.085 | 0.167 | 0.065 | 0.085 | 0.047 | 0.091 |
| FreTS | **0.050** | **0.076** | **0.080** | **0.166** | **0.062** | **0.082** | **0.046** | **0.090** |

nel learner plays a more important role in short-term forecasting. In long-term forecasting, we note that the frequency temporal learner is more effective than the frequency channel learner. In Appendix E.1, we also conduct the experiments and report performance on other datasets. Interestingly, we find out the channel learner would lead to the worse performance in some long-term forecasting cases. A potential explanation is that the channel independent strategy [39] brings more benefit to forecasting.

**FreMLP vs. MLP** We further study the effectiveness of FreMLP in time series forecasting. We use FreMLP to replace the original MLP component in the existing SOTA MLP-based models (i.e., DLinear and NLinear [35]), and compare their performances with the original DLinear and NLinear under the same experimental settings. The experimental results are presented in Table 4. From the table, we easily observe that for any prediction length, the performance of both DLinear and NLinear models has been improved after replacing the corresponding MLP component with our FreMLP. Quantitatively, incorporating FreMLP into the DLinear model brings an average improvement of 6.4% in MAE and 11.4% in RMSE on the Exchange dataset, and 4.9% in MAE and 3.5% in RMSE on the Weather dataset. A similar improvement has also been achieved on the two datasets with regard to NLinear, according to Table 4. These results confirm the effectiveness of FreMLP compared to MLP again and we include more implementation details and analysis in Appendix B.5.

Table 4: Ablation study on the Exchange and Weather datasets with a lookback window size of 96 and the prediction length $\tau \in \{96, 192, 336, 720\}$. DLinear (FreMLP)/NLinear (FreMLP) means that we replace the MLPs in DLinear/NLinear with FreMLP. The best results are in **bold**.

| Datasets | Exchange | | | | | | | | Weather | | | | | | | |
|---|---|---|---|---|---|---|---|---|---|---|---|---|---|---|---|---|
| Lengths | 96 | | 192 | | 336 | | 720 | | 96 | | 192 | | 336 | | 720 | |
| Metrics | MAE | RMSE | MAE | RMSE | MAE | RMSE | MAE | RMSE | MAE | RMSE | MAE | RMSE | MAE | RMSE | MAE | RMSE |
| DLinear | 0.037 | 0.051 | 0.054 | 0.072 | 0.071 | 0.095 | 0.095 | 0.119 | 0.041 | 0.081 | 0.047 | 0.089 | 0.056 | 0.098 | 0.065 | 0.106 |
| DLinear (FreMLP) | **0.036** | **0.049** | **0.053** | **0.071** | **0.063** | **0.071** | **0.086** | **0.101** | **0.038** | **0.078** | **0.045** | **0.086** | **0.055** | **0.097** | **0.061** | **0.100** |
| NLinear | 0.037 | 0.051 | 0.051 | 0.069 | 0.069 | 0.093 | 0.115 | 0.146 | 0.037 | 0.081 | 0.045 | 0.089 | 0.052 | 0.098 | 0.058 | 0.106 |
| NLinear (FreMLP) | **0.036** | **0.050** | **0.049** | **0.067** | **0.067** | **0.091** | **0.109** | **0.139** | **0.035** | **0.076** | **0.043** | **0.084** | **0.050** | **0.094** | **0.057** | **0.103** |

## 4.3 Efficiency Analysis

The complexity of our proposed FreTS is $\mathcal{O}(N \log N + L \log L)$. We perform efficiency comparisons with some state-of-the-art GNN-based methods and Transformer-based models under different numbers of variables $N$ and prediction lengths $\tau$, respectively. On the Wiki dataset, we conduct experiments over $N \in \{1000, 2000, 3000, 4000, 5000\}$ under the same lookback window size of 12

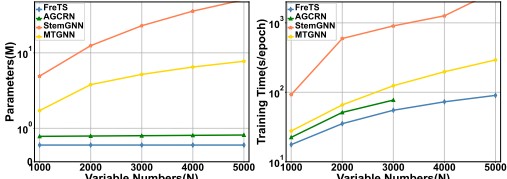
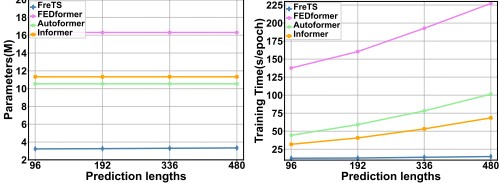

(a) Parameters (left) and training time (right) under different variable numbers

(b) Parameters (left) and training time (right) under different prediction lengths

Figure 4: Efficiency analysis (model parameters and training time) on the Wiki and Exchange dataset. (a) The efficiency comparison under different number of variables: the number of variables is enlarged from 1000 to 5000 with the input window size as 12 and the prediction length as 12 on Wiki dataset. (b) The efficiency comparison under the prediction lengths: we conduct experiments with prediction lengths prolonged from 96 to 480 under the same window size of 96 on the Exchange dataset.

and prediction length of 12, as shown in Figure 4(a). From the figure, we can find that: (1) The amount of FreTS parameters is agnostic to $N$. (2) Compared with AGCRN, FreTS incurs an average 30% reduction of the number of parameters and 20% reduction of training time. On the Exchange dataset, we conduct experiments on different prediction lengths $\tau \in \{96, 192, 336, 480\}$ with the same input length of 96. The results are shown in Figure 4(b). It demonstrates: (1) Compared with Transformer-based methods (FEDformer [30], Autoformer [14], and Informer [13]), FreTS reduces the number of parameters by at least 3 times. (2) The training time of FreTS is averagely 3 times faster than Informer, 5 times faster than Autoformer, and more than 10 times faster than FEDformer. These show our great potential in real-world deployment.

## 4.4 Visualization Analysis

In Figure 5, we visualize the learned weights $\mathcal{W}$ in FreMLP on the Traffic dataset with a lookback window size of 48 and prediction length of 192. As the weights $\mathcal{W}$ are complex numbers, we provide visualizations of the real part $\mathcal{W}_r$ (presented in (a)) and the imaginary part $\mathcal{W}_i$ (presented in (b)) separately. From the figure, we can observe that both the real and imaginary parts play a crucial role in learning process: the weight coefficients of the real or imaginary part exhibit energy aggregation characteristics (clear diagonal patterns) which can facilitate to learn the significant features. In Appendix E.2, we further conduct a

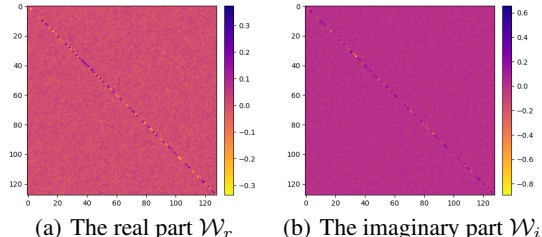

(a) The real part $\mathcal{W}_r$     (b) The imaginary part $\mathcal{W}_i$

Figure 5: Visualizing learned weights of FreMLP on the Traffic dataset. $\mathcal{W}_r$ represents the real part of $\mathcal{W}$, and $\mathcal{W}_i$ represents the imaginary part.

detailed analysis on the effects of the real and imaginary parts in different contexts of forecasting, and the effects of the two parts in the FreMLP. We examine their individual contributions and investigate how they influence the final performance. Additional visualizations of the weights on different datasets with various settings, as well as visualizations of global periodic patterns, can be found in Appendix G.1 and Appendix G.2, respectively.

## 5 Conclusion Remarks

In this paper, we explore a novel direction and make a new attempt to apply frequency-domain MLPs for time series forecasting. We have redesigned MLPs in the frequency domain that can effectively capture the underlying patterns of time series with global view and energy compaction. We then verify this design by a simple yet effective architecture, FreTS, built upon the frequency-domain MLPs for time series forecasting. Our comprehensive empirical experiments on seven benchmarks of short-term forecasting and six benchmarks of long-term forecasting have validated the superiority of our proposed methods. Simple MLPs have several advantages and lay the foundation of modern deep learning, which have great potential for satisfied performance with high efficiency. We hope this work can facilitate more future research of MLPs on time series modeling.

## Acknowledgments and Disclosure of Funding

The work was supported in part by the National Key Research and Development Program of China under Grant 2020AAA0104903 and 2019YFB1406300, and National Natural Science Foundation of China under Grant 62072039 and 62272048.

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

# A Notations

Table 5: Notation.

| | |
|---|---|
| $\mathbf{X}_t$ | multivariate time series with a lookback window of $L$ at timestamps t, $\mathbf{X}_t \in \mathbb{R}^{N \times L}$ |
| $X_t$ | the multivariate values of $N$ distinct series at timestamp $t$, $X_t \in \mathbb{R}^N$ |
| $\mathbf{Y}_t$ | the prediction target with a horizon window of length $\tau$ at timestamps $t$, $\mathbf{Y}_t \in \mathbb{R}^{N \times \tau}$ |
| $\mathbf{H}_t$ | the hidden representation of $\mathbf{X}_t$, $\mathbf{H}_t \in \mathbb{R}^{N \times L \times d}$ |
| $\mathbf{Z}_t$ | the output of the frequency channel learner, $\mathbf{Z}_t \in \mathbb{R}^{N \times L \times d}$ |
| $\mathbf{S}_t$ | the output of the frequency temporal learner, $\mathbf{S}_t \in \mathbb{R}^{N \times L \times d}$ |
| $\mathcal{H}_{chan}$ | the domain conversion of $\mathbf{H}_t$ on channel dimensions, $\mathcal{H}_{chan} \in \mathbb{C}^{N \times L \times d}$ |
| $\mathcal{Z}_{chan}$ | the FreMLP output of $\mathcal{H}_{chan}$, $\mathcal{Z}_{chan} \in \mathbb{C}^{N \times L \times d}$ |
| $\mathcal{Z}_{temp}$ | the domain conversion of $\mathbf{Z}_t$ on temporal dimensions, $\mathcal{Z}_{temp} \in \mathbb{C}^{N \times L \times d}$ |
| $\mathcal{S}_{temp}$ | the FreMLP output of $\mathcal{Z}_{temp}$, $\mathcal{S}_{temp} \in \mathbb{C}^{N \times L \times d}$ |
| $\mathcal{W}^{chan}$ | the complex number weight matrix of FreMLP in the frequency channel learner, $\mathcal{W}^{chan} \in \mathbb{C}^{d \times d}$ |
| $\mathcal{B}^{chan}$ | the complex number bias of FreMLP in the frequency channel learner, $\mathcal{B}^{chan} \in \mathbb{C}^{d}$ |
| $\mathcal{W}^{temp}$ | the complex number weight matrix of FreMLP in the frequency temporal learner, $\mathcal{W}^{temp} \in \mathbb{C}^{d \times d}$ |
| $\mathcal{B}^{temp}$ | the complex number bias of FreMLP in the frequency temporal learner, $\mathcal{B}^{temp} \in \mathbb{C}^{d}$ |

# B Experimental Details

## B.1 Datasets

We adopt thirteen real-world benchmarks in the experiments to evaluate the accuracy of short-term and long-term forecasting. The details of the datasets are as follows:

**Solar**[4]: It is about the solar power collected by National Renewable Energy Laboratory. We choose the power plant data points in Florida as the data set which contains 593 points. The data is collected from 01/01/2006 to 31/12/2016 with the sampling interval of every 1 hour.

**Wiki [37]**: It contains a number of daily views of different Wikipedia articles and is collected from 1/7/2015 to 31/12/2016. It consists of approximately $145k$ time series and we randomly choose $5k$ from them as our experimental data set.

**Traffic [37]**: It contains hourly traffic data from 963 San Francisco freeway car lanes for short-term forecasting settings while it contains 862 car lanes for long-term forecasting. It is collected since 01/01/2015 with a sampling interval of every 1 hour.

**ECG**[5]: It is about Electrocardiogram(ECG) from the UCR time-series classification archive. It contains 140 nodes and each node has a length of 5000.

---

[4]https://www.nrel.gov/grid/solar-power-data.html
[5]http://www.timeseriesclassification.com/description.php?Dataset=ECG5000

**Electricity**[6]: It contains electricity consumption of 370 clients for short-term forecasting while it contains electricity consumption of 321 clients for long-term forecasting. It is collected since 01/01/2011. The data sampling interval is every 15 minutes.

**COVID-19 [4]**: It is about COVID-19 hospitalization in the U.S. state of California (CA) from 01/02/2020 to 31/12/2020 provided by the Johns Hopkins University with the sampling interval of every day.

**METR-LA**[7]: It contains traffic information collected from loop detectors in the highway of Los Angeles County. It contains 207 sensors which are from 01/03/2012 to 30/06/2012 and the data sampling interval is every 5 minutes.

**Exchange**[8]: It contains the collection of the daily exchange rates of eight foreign countries including Australia, British, Canada, Switzerland, China, Japan, New Zealand, and Singapore ranging from 1990 to 2016 and the data sampling interval is every 1 day.

**Weather**[9]: It collects 21 meteorological indicators, such as humidity and air temperature, from the Weather Station of the Max Planck Biogeochemistry Institute in Germany in 2020. The data sampling interval is every 10 minutes.

**ETT**[10]: It is collected from two different electric transformers labeled with 1 and 2, and each of them contains 2 different resolutions (15 minutes and 1 hour) denoted with m and h. We use ETTh1 and ETTm1 as our long-term forecasting benchmarks.

## B.2 Baselines

We adopt eighteen representative and state-of-the-art baselines for comparison including LSTM-based models, GNN-based models, and Transformer-based models. We introduce these models as follows:

**VAR** [23]: VAR is a classic linear autoregressive model. We use the Statsmodels library (`https://www.statsmodels.org`) which is a Python package that provides statistical computations to realize the VAR.

**DeepGLO** [37]: DeepGLO models the relationships among variables by matrix factorization and employs a temporal convolution neural network to introduce non-linear relationships. We download the source code from: `https://github.com/rajatsen91/deepglo`. We use the recommended configuration as our experimental settings for Wiki, Electricity, and Traffic datasets. For the COVID-19 dataset, the vertical and horizontal batch size is set to 64, the rank of the global model is set to 64, the number of channels is set to [32, 32, 32, 1], and the period is set to 7.

**LSTNet** [10]: LSTNet uses a CNN to capture inter-variable relationships and an RNN to discover long-term patterns. We download the source code from: `https://github.com/laiguokun/LSTNet`. In our experiment, we use the recommended configuration where the number of CNN hidden units is 100, the kernel size of the CNN layers is 4, the dropout is 0.2, the RNN hidden units is 100, the number of RNN hidden layers is 1, the learning rate is 0.001 and the optimizer is Adam.

**TCN** [11]: TCN is a causal convolution model for regression prediction. We download the source code from: `https://github.com/locuslab/TCN`. We utilize the same configuration as the polyphonic music task exampled in the open source code where the dropout is 0.25, the kernel size is 5, the number of hidden units is 150, the number of levels is 4 and the optimizer is Adam.

**Informer** [13]: Informer leverages an efficient self-attention mechanism to encode the dependencies among variables. We download the source code from: `https://github.com/zhouhaoyi/Informer2020`. We use the recommended configuration as the experimental settings where the dropout is 0.05, the number of encoder layers is 2, the number of decoder layers is 1, the learning rate is 0.0001, and the optimizer is Adam.

**Reformer** [18]: Reformer combines the modeling capacity of a Transformer with an architecture that can be executed efficiently on long sequences and with small memory use. We download the source

---

[6]`https://archive.ics.uci.edu/ml/datasets/ElectricityLoadDiagrams20112014`

[7]`https://github.com/liyaguang/DCRNN`

[8]`https://github.com/laiguokun/multivariate-time-series-data`

[9]`https://www.bgc-jena.mpg.de/wetter/`

[10]`https://github.com/zhouhaoyi/ETDataset`

code from: `https://github.com/thuml/Autoformer`. We use the recommended configuration as the experimental settings.

**Autoformer** [14]: Autoformer proposes a decomposition architecture by embedding the series decomposition block as an inner operator, which can progressively aggregate the long-term trend part from intermediate prediction. We download the source code from: `https://github.com/thuml/Autoformer`. We use the recommended configuration as the experimental settings.

**FEDformer** [30]: FEDformer proposes an attention mechanism with low-rank approximation in frequency and a mixture of expert decomposition to control the distribution shifting. We download the source code from: `https://github.com/MAZiqing/FEDformer`. We use FEB-f as the Frequency Enhanced Block and select the random mode with 64 as the experimental mode.

**SFM** [29]: On the basis of the LSTM model, SFM introduces a series of different frequency components in the cell states. We download the source code from: `https://github.com/z331565360/State-Frequency-Memory-stock-prediction`. We follow the recommended configuration as the experimental settings where the learning rate is 0.01, the frequency dimension is 10, the hidden dimension is 10 and the optimizer is RMSProp.

**StemGNN** [16]: StemGNN leverages GFT and DFT to capture dependencies among variables in the frequency domain. We download the source code from: `https://github.com/microsoft/StemGNN`. We use the recommended configuration of stemGNN as our experiment setting where the optimizer is RMSProp, the learning rate is 0.0001, the number of stacked layers is 5, and the dropout rate is 0.5.

**MTGNN** [15]: MTGNN proposes an effective method to exploit the inherent dependency relationships among multiple time series. We download the source code from: `https://github.com/nnzhan/MTGNN`. Because the experimental datasets have no static features, we set the parameter load_static_feature to false. We construct the graph by the adaptive adjacency matrix and add the graph convolution layer. Regarding other parameters, we follow the recommended settings.

**GraphWaveNet** [27]: GraphWaveNet introduces an adaptive dependency matrix learning to capture the hidden spatial dependency. We download the source code from: `https://github.com/nnzhan/Graph-WaveNet`. Since our datasets have no prior defined graph structures, we use only adaptive adjacent matrix. We add a graph convolutional layer and randomly initialize the adjacent matrix. We adopt the recommended setting as its experimental configuration where the learning rate is 0.001, the dropout is 0.3, the number of epochs is 50, and the optimizer is Adam.

**AGCRN** [17]: AGCRN proposes a data-adaptive graph generation module for discovering spatial correlations from data. We download the source code from: `https://github.com/LeiBAI/AGCRN`. We follow the recommended settings where the embedding dimension is 10, the learning rate is 0.003, and the optimizer is Adam.

**TAMP-S2GCNets** [4]: TAMP-S2GCNets explores the utility of MP to enhance knowledge representation mechanisms within the time-aware DL paradigm. We download the source code from: `https://www.dropbox.com/sh/n0ajd5l0tdeyb80/AABGn-ejfV1YtRwjf_LOAOsNa?dl=0`. TAMP-S2GCNets require a pre-defined graph topology and we use the California State topology provided by the source code as input. We adopt the recommended settings as the experimental configuration for COVID-19.

**DCRNN** [38]: DCRNN uses bidirectional graph random walk to model spatial dependency and recurrent neural network to capture the temporal dynamics. We download the source code from: `https://github.com/liyaguang/DCRNN`. We use the recommended configuration as our experimental settings with the batch size is 64, the learning rate is $0.01$, the input dimension is 2 and the optimizer is Adam. DCRNN requires a pre-defined graph structure and we use the adjacency matrix as the pre-defined structure provided by the METR-LA dataset.

**STGCN** [40]: STGCN integrates graph convolution and gated temporal convolution through spatial-temporal convolutional blocks. We download the source code from: `https://github.com/VeritasYin/STGCN_IJCAI-18`. We follow the recommended settings as our experimental configuration where the batch size is 50, the learning rate is $0.001$ and the optimizer is Adam. STGCN requires a pre-defined graph structure and we leverage the adjacency matrix as the pre-defined structure provided by the METR-LA dataset.

**LTSF-Linear** [35]: LTSF-Linear proposes a set of embarrassingly simple one-layer linear models to

learn temporal relationships between input and output sequences. We download the source code from: `https://github.com/cure-lab/LTSF-Linear`. We use it as our long-term forecasting baseline and follow the recommended settings as experimental configuration.

**PatchTST** [39]: PatchTST proposes an effective design of Transformer-based models for time series forecasting tasks by introducing two key components: patching and channel-independent structure. We download the source code from: `https://github.com/PatchTST`. We use it as our long-term forecasting baseline and adhere to the recommended settings as the experimental configuration.

### B.3 Implementation Details

By default, both the frequency channel and temporal learners contain one layer of $\mathrm{FreMLP}$ with the embedding size $d$ of 128, and the hidden size $d_h$ is set to 256. For short-term forecasting, the batch size is set to 32 for Solar, METR-LA, ECG, COVID-19, and Electricity datasets. And for Wiki and Traffic datasets, the batch size is set to 4. For the long-term forecasting, except for the lookback window size, we follow most of the experimental settings of LTSF-Linear [35]. The lookback window size is set to 96 which is recommended by FEDformer [30] and Autoformer [14]. In Appendix F.2, we also use 192 and 336 as the lookback window size to conduct experiments and the results demonstrate that FreTS outperforms other baselines as well. For the longer prediction lengths (e.g., 336, 720), we use the channel independence strategy and contain only the frequency temporal learner in our model. For some datasets, we carefully tune the hyperparameters including the batch size and learning rate on the validation set, and we choose the settings with the best performance. We tune the batch size over {4, 8, 16, 32}.

### B.4 Visualization Settings

**The Visualization Method for Global View**. We follow the visualization methods in LTSF-Linear [35] to visualize the weights learned in the time domain on the input (corresponding to the left side of Figure 1(a)). For the visualization of the weights learned on the frequency spectrum, we first transform the input into the frequency domain and select the real part of the input frequency spectrum to replace the original input. Then, we learn the weights and visualize them in the same manner as in the time domain. The right side of Figure 1(a) shows the weights learned on the Traffic dataset with a lookback window of 96 and a prediction length of 96, Figure 9 displays the weights learned on the Traffic dataset with a lookback window of 72 and a prediction length of 336, and Figure 10 is the weights learned on the Electricity dataset with a lookback window of 96 and a prediction length of 96.

**The Visualization Method for Energy Compaction**. Since the learned weights $\mathcal{W} = \mathcal{W}_r + j\mathcal{W}_i \in \mathbb{C}^{d \times d}$ of the frequency-domain MLPs are complex numbers, we visualize the corresponding real part $\mathcal{W}_r$ and imaginary part $\mathcal{W}_i$, respectively. We normalize them by the calculation of $1/\max(\mathcal{W}) * \mathcal{W}$ and visualize the normalization values. The right side of Figure 1(b) is the real part of $\mathcal{W}$ learned on the Traffic dataset with a lookback window of 48 and a prediction length of 192. To visualize the corresponding weights learned in the time domain, we replace the frequency spectrum of input $\mathcal{Z}_{temp} \in \mathbb{C}^{N \times L \times d}$ with the original time domain input $\mathbf{H}_t \in \mathbb{R}^{N \times L \times d}$ and perform calculations in the time domain with a weight $W \in \mathbb{R}^{d \times d}$, as depicted in the left side of Figure 1(b).

### B.5 Ablation Experimental Settings

DLinear decomposes a raw data input into a trend component and a seasonal component, and two one-layer linear layers are applied to each component. In the ablation study part, we replace the two linear layers with two different frequency-domain MLPs (corresponding to DLinear (FreMLP) in Table 4), and compare their accuracy using the same experimental settings recommended in LTSF-Linear [35]. NLinear subtracts the input by the last value of the sequence. Then, the input goes through a linear layer, and the subtracted part is added back before making the final prediction. We replace the linear layer with a frequency-domain MLP (corresponding to NLinear (FreMLP) in Table 4), and compare their accuracy using the same experimental settings recommended in LTSF-Linear [35].

## C Complex Multiplication

For two complex number values $\mathcal{Z}_1 = (a + jb)$ and $\mathcal{Z}_2 = (c + jd)$, where $a$ and $c$ is the real part of $\mathcal{Z}_1$ and $\mathcal{Z}_2$ respectively, $b$ and $d$ is the imaginary part of $\mathcal{Z}_1$ and $\mathcal{Z}_2$ respectively. Then the multiplication of $\mathcal{Z}_1$ and $\mathcal{Z}_2$ is calculated by:

$$\mathcal{Z}_1 \mathcal{Z}_2 = (a + jb)(c + jd) = ac + j^2 bd + jad + jbc = (ac - bd) + j(ad + bc) \qquad (10)$$

where $j^2 = -1$.

## D Proof

### D.1 Proof of Theorem 1

**Theorem 1.** *Suppose that $\mathbf{H}$ is the representation of raw time series and $\mathcal{H}$ is the corresponding frequency components of the spectrum, then the energy of a time series in the time domain is equal to the energy of its representation in the frequency domain. Formally, we can express this with above notations by:*

$$\int_{-\infty}^{\infty} |\mathbf{H}(v)|^2 \mathrm{d}v = \int_{-\infty}^{\infty} |\mathcal{H}(f)|^2 \mathrm{d}f \qquad (11)$$

*where $\mathcal{H}(f) = \int_{-\infty}^{\infty} \mathbf{H}(v)e^{-j2\pi fv}\mathrm{d}v$, $v$ is the time/channel dimension, $f$ is the frequency dimension.*

*Proof.* Given the representation of raw time series $\mathbf{H} \in \mathbb{R}^{N \times L \times d}$, let us consider performing integration in either the $N$ dimension (channel dimension) or the $L$ dimension (temporal dimension), denoted as the integral over $v$, then

$$\int_{-\infty}^{\infty} |\mathbf{H}(v)|^2 \mathrm{d}v = \int_{-\infty}^{\infty} \mathbf{H}(v)\mathbf{H}^*(v)\mathrm{d}v$$

where $\mathbf{H}^*(v)$ is the conjugate of $\mathbf{H}(v)$. According to IDFT, $\mathbf{H}^*(v) = \int_{-\infty}^{\infty} \mathcal{H}^*(f)e^{-j2\pi fv}\mathrm{d}f$, we can obtain

$$\begin{aligned}
\int_{-\infty}^{\infty} |\mathbf{H}(v)|^2 \mathrm{d}v &= \int_{-\infty}^{\infty} \mathbf{H}(v)[\int_{-\infty}^{\infty} \mathcal{H}^*(f)e^{-j2\pi fv}\mathrm{d}f]\mathrm{d}v \\
&= \int_{-\infty}^{\infty} \mathcal{H}^*(f)[\int_{-\infty}^{\infty} \mathbf{H}(v)e^{-j2\pi fv}\mathrm{d}v]\mathrm{d}f \\
&= \int_{-\infty}^{\infty} \mathcal{H}^*(f)\mathcal{H}(f)\mathrm{d}f \\
&= \int_{-\infty}^{\infty} |\mathcal{H}(f)|^2 \mathrm{d}f
\end{aligned}$$

Proved. $\square$

Therefore, the energy of a time series in the time domain is equal to the energy of its representation in the frequency domain.

### D.2 Proof of Theorem 2

**Theorem 2.** *Given the time series input $\mathbf{H}$ and its corresponding frequency domain conversion $\mathcal{H}$, the operations of frequency-domain MLP on $\mathcal{H}$ can be represented as global convolutions on $\mathbf{H}$ in the time domain. This can be given by:*

$$\mathcal{H}\mathcal{W} + \mathcal{B} = \mathcal{F}(\mathbf{H} * W + B) \qquad (12)$$

*where $*$ is a circular convolution, $\mathcal{W}$ and $\mathcal{B}$ are the complex number weight and bias, $W$ and $B$ are the weight and bias in the time domain, and $\mathcal{F}$ is DFT.*

*Proof.* Suppose that we conduct operations in the $N$ (i.e., channel dimension) or $L$ (i.e., temporal dimension) dimension, then

$$\mathcal{F}(\mathbf{H}(v) * W(v)) = \int_{-\infty}^{\infty} (\mathbf{H}(v) * W(v))e^{-j2\pi fv}\mathrm{d}v$$

According to convolution theorem, $\mathbf{H}(v) * W(v) = \int_{-\infty}^{\infty}(\mathbf{H}(\tau)W(v-\tau))\mathrm{d}\tau$, then

$$\mathcal{F}(\mathbf{H}(v) * W(v)) = \int_{-\infty}^{\infty} \int_{-\infty}^{\infty} (\mathbf{H}(\tau)W(v-\tau))e^{-j2\pi fv}\mathrm{d}\tau\mathrm{d}v$$

$$= \int_{-\infty}^{\infty} \int_{-\infty}^{\infty} W(v-\tau)e^{-j2\pi fv}\mathrm{d}v\mathbf{H}(\tau)\mathrm{d}\tau$$

Let $x = v - \tau$, then

$$\mathcal{F}(\mathbf{H}(v) * W(v)) = \int_{-\infty}^{\infty} \int_{-\infty}^{\infty} W(x)e^{-j2\pi f(x+\tau)}\mathrm{d}x\mathbf{H}(\tau)\mathrm{d}\tau$$

$$= \int_{-\infty}^{\infty} \int_{-\infty}^{\infty} W(x)e^{-j2\pi fx}e^{-j2\pi f\tau}\mathrm{d}x\mathbf{H}(\tau)\mathrm{d}\tau$$

$$= \int_{-\infty}^{\infty} \mathbf{H}(\tau)e^{-j2\pi f\tau}\mathrm{d}\tau \int_{-\infty}^{\infty} W(x)e^{-j2\pi fx}\mathrm{d}x$$

$$= \mathcal{H}(f)\mathcal{W}(f)$$

Accordingly, $(\mathbf{H}(v) * W(v))$ in the time domain is equal to $(\mathcal{H}(f)\mathcal{W}(f))$ in the frequency domain. Therefore, the operations of $\mathrm{FreMLP}$ ($\mathcal{HW} + \mathcal{B}$) in the channel (i.e., $v = N$) or temporal dimension (i.e., $v = L$), are equal to the operations ($\mathbf{H} * W + B$) in the time domain. This implies that frequency-domain MLPs can be viewed as global convolutions in the time domain. Proved. □

## E  Further Analysis

### E.1  Ablation Study

In this section, we further analyze the effects of the frequency channel and temporal learners with different prediction lengths on ETTm1 and ETTh1 datasets. The results are shown in Table 6. It demonstrates that with the prediction length increasing, the frequency temporal learner shows more effective than the channel learner. Especially, when the prediction length is longer (e.g., 336, 720), the channel learner will lead to worse performance. The reason is that when the prediction lengths become longer, the model with the channel learner is likely to overfit data during training. Thus for long-term forecasting with longer prediction lengths, the channel independence strategy may be more effective, as described in PatchTST [39].

Table 6: Ablation studies of the frequency channel and temporal learners in long-term forecasting. 'I/O' indicates lookback window sizes/prediction lengths.

| Dataset | ETTm1 | | | | | | | | ETTh1 | | | | | | | |
|---------|-------|-------|-------|-------|-------|-------|-------|-------|-------|-------|-------|-------|-------|-------|-------|-------|
| I/O | 96/96 | | 96/192 | | 96/336 | | 96/720 | | 96/96 | | 96/192 | | 96/336 | | 96/720 | |
| Metrics | MAE | RMSE | MAE | RMSE | MAE | RMSE | MAE | RMSE | MAE | RMSE | MAE | RMSE | MAE | RMSE | MAE | RMSE |
| FreCL | 0.053 | 0.078 | 0.059 | 0.085 | 0.067 | 0.095 | 0.097 | 0.125 | 0.063 | 0.089 | 0.067 | 0.093 | 0.071 | 0.097 | 0.087 | 0.115 |
| FreTL | 0.053 | 0.078 | 0.058 | 0.084 | **0.062** | **0.089** | **0.069** | **0.096** | **0.061** | **0.087** | 0.065 | **0.091** | **0.070** | **0.096** | **0.082** | **0.108** |
| FreTS | **0.052** | **0.077** | **0.057** | **0.083** | 0.064 | 0.092 | 0.071 | 0.099 | 0.063 | 0.089 | 0.066 | 0.092 | 0.072 | 0.098 | 0.086 | 0.113 |

### E.2  Impacts of Real/Imaginary Parts

To investigate the effects of real and imaginary parts, we conduct experiments on Exchange and ETTh1 datasets under different prediction lengths $L \in \{96, 192\}$ with the lookback window of 96. Furthermore, we analyze the effects of $\mathcal{W}_r$ and $\mathcal{W}_i$ in the weights $\mathcal{W} = \mathcal{W}_r + j\mathcal{W}_i$ of $\mathrm{FreMLP}$. In this experiment, we only use the frequency temporal learner in our model. The results are shown in

Table 7. In the table, Input$_{real}$ indicates that we only feed the real part of the input into the network, and Input$_{imag}$ indicates that we only feed the imaginary part of the input into the network. $\mathcal{W}(\mathcal{W}_r)$ denotes that we set $\mathcal{W}_i$ to 0 and $\mathcal{W}(\mathcal{W}_i)$ denotes that we set $\mathcal{W}_r$ to 0. From the table, we can observe that both the real part and imaginary part of input are indispensable and the real part is more important to the imaginary part, and the real part of $\mathcal{W}$ plays a more significant role for the model performances.

Table 7: Investigation the impacts of real/imaginary parts

| Dataset | Exchange | | | | ETTh1 | | | |
|---|---|---|---|---|---|---|---|---|
| I/O | 96/96 | | 96/192 | | 96/96 | | 96/192 | |
| Metrics | MAE | RMSE | MAE | RMSE | MAE | RMSE | MAE | RMSE |
| Input$_{real}$ | 0.048 | 0.062 | 0.058 | 0.074 | 0.080 | 0.111 | 0.083 | 0.113 |
| Input$_{imag}$ | 0.143 | 0.185 | 0.143 | 0.184 | 0.130 | 0.156 | 0.130 | 0.156 |
| $\mathcal{W}(\mathcal{W}_r)$ | 0.039 | 0.053 | 0.051 | 0.067 | 0.063 | 0.089 | 0.067 | 0.093 |
| $\mathcal{W}(\mathcal{W}_i)$ | 0.143 | 0.184 | 0.142 | 0.184 | 0.116 | 0.138 | 0.117 | 0.139 |
| FreTS | 0.037 | 0.051 | 0.050 | 0.067 | 0.061 | 0.087 | 0.065 | 0.091 |

## E.3 Parameter Sensitivity

We further perform extensive experiments on the ECG dataset to evaluate the sensitivity of the input length $L$ and the embedding dimension size $d$. (1) *Input length*: We tune over the input length with the value $\{6, 12, 18, 24, 30, 36, 42, 50, 60\}$ on the ECG dataset and the prediction length is 12, and the result is shown in Figure 6(a). From the figure, we can find that with the input length increasing, the performance first becomes better because the long input length may contain more pattern information, and then it decreases due to data redundancy or overfitting. (2) *Embedding size*: We choose the embedding size over the set $\{32, 64, 128, 256, 512\}$ on the ECG dataset. The results are shown in Figure 6(b). It shows that the performance first increases and then decreases with the increase of the embedding size because a large embedding size improves the fitting ability of our FreTS but may easily lead to overfitting especially when the embedding size is too large.

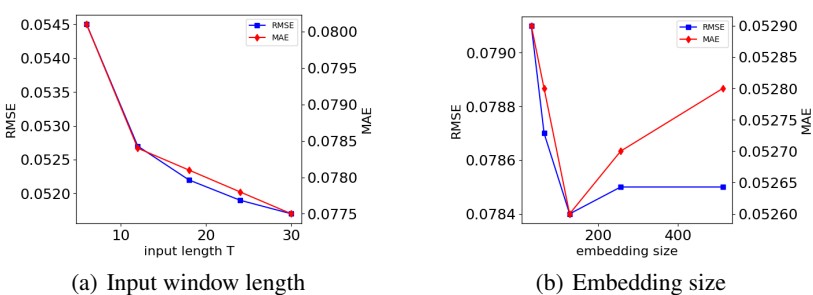

(a) Input window length          (b) Embedding size

Figure 6: The parameter sensitivity analyses of FreTS.

## F  Additional Results

### F.1  Multi-Step Forecasting

To further evaluate the performance of our FreTS in multi-step forecasting, we conduct more experiments on METR-LA and COVID-19 datasets with the input length of 12 and the prediction lengths of $\{3, 6, 9, 12\}$, and the results are shown in Tables 8 and 9, respectively. In this experiment, we only select the state-of-the-art (i.e., GNN-based and Transformer-based) models as the baselines since they perform better than other models, such as RNN and TCN. Among these baselines, STGCN, DCRNN, and TAMP-S2GCNets require pre-defined graph structures. The results demonstrate that

FreTS outperforms other baselines, including those models with pre-defined graph structures, at all steps. This further confirms that FreTS has strong capabilities in capturing channel-wise and time-wise dependencies.

Table 8: Multi-step short-term forecasting results comparison on the METR-LA dataset with the input length of 12 and the prediction length of $\tau \in \{3, 6, 9, 12\}$. We highlight the best results in **bold** and the second best results are underline.

| Length | 3 | | 6 | | 9 | | 12 | |
|---|---|---|---|---|---|---|---|---|
| Metrics | MAE | RMSE | MAE | RMSE | MAE | RMSE | MAE | RMSE |
| Reformer | 0.086 | 0.154 | 0.097 | 0.176 | 0.107 | 0.193 | 0.118 | 0.206 |
| Informer | 0.082 | 0.156 | 0.094 | 0.176 | 0.108 | 0.193 | 0.125 | 0.214 |
| Autoformer | 0.087 | 0.149 | 0.091 | 0.162 | 0.106 | 0.178 | 0.099 | 0.184 |
| FEDformer | 0.064 | 0.127 | 0.073 | 0.145 | 0.079 | 0.160 | 0.086 | 0.175 |
| DCRNN | 0.160 | 0.204 | 0.191 | 0.243 | 0.216 | 0.269 | 0.241 | 0.291 |
| STGCN | 0.058 | 0.133 | 0.080 | 0.177 | 0.102 | 0.209 | 0.128 | 0.238 |
| GraphWaveNet | 0.180 | 0.366 | 0.184 | 0.375 | 0.196 | 0.382 | 0.202 | 0.386 |
| MTGNN | 0.135 | 0.294 | 0.144 | 0.307 | 0.149 | 0.328 | 0.153 | 0.316 |
| StemGNN | 0.052 | 0.115 | 0.069 | 0.141 | 0.080 | 0.162 | 0.093 | 0.175 |
| AGCRN | 0.062 | 0.131 | 0.086 | 0.165 | 0.099 | 0.188 | 0.109 | 0.204 |
| **FreTS** | **0.050** | **0.113** | **0.066** | **0.140** | **0.076** | **0.158** | **0.080** | **0.166** |

Table 9: Multi-step short-term forecasting results comparison on the COVID-19 dataset with the input length of 12 and the prediction length of $\tau \in \{3, 6, 9, 12\}$. We highlight the best results in **bold** and the second best results are underline.

| Length | 3 | | 6 | | 9 | | 12 | |
|---|---|---|---|---|---|---|---|---|
| Metrics | MAE | RMSE | MAE | RMSE | MAE | RMSE | MAE | RMSE |
| Reformer | 0.212 | 0.282 | 0.139 | 0.186 | 0.148 | 0.197 | 0.152 | 0.209 |
| Informer | 0.234 | 0.312 | 0.190 | 0.245 | 0.184 | 0.242 | 0.200 | 0.259 |
| Autoformer | 0.212 | 0.280 | 0.144 | 0.191 | 0.152 | 0.201 | 0.159 | 0.211 |
| FEDformer | 0.246 | 0.328 | 0.169 | 0.242 | 0.175 | 0.247 | 0.160 | 0.219 |
| GraphWaveNet | 0.092 | 0.129 | 0.133 | 0.179 | 0.171 | 0.225 | 0.201 | 0.255 |
| StemGNN | 0.247 | 0.318 | 0.344 | 0.429 | 0.359 | 0.442 | 0.421 | 0.508 |
| AGCRN | 0.130 | 0.172 | 0.171 | 0.218 | 0.224 | 0.277 | 0.254 | 0.309 |
| MTGNN | 0.276 | 0.379 | 0.446 | 0.513 | 0.484 | 0.548 | 0.394 | 0.488 |
| TAMP-S2GCNets | 0.140 | 0.190 | 0.150 | 0.200 | 0.170 | 0.230 | 0.180 | 0.230 |
| **FreTS** | **0.071** | **0.103** | **0.093** | **0.131** | **0.109** | **0.148** | **0.124** | **0.164** |

### F.2 Long-Term Forecasting under Varying Lookback Window

In Table 10, we present the long-term forecasting results of our FreTS and other baselines (PatchTST [39], LTSF-linear [35], FEDformer [30], Autoformer [14], Informer [13], and Reformer [18]) under different lookback window lengths $L \in \{96, 192, 336\}$ on the Exchange dataset. The prediction lengths are $\{96, 192, 336, 720\}$. From the table, we can observe that our FreTS outperforms all baselines in all settings and achieves significant improvements than FEDformer [30], Autoformer [14], Informer [13], and Reformer [18]. It verifies the effectiveness of our FreTS in learning informative representation under different lookback window.

## G   Visualizations

### G.1   Weight Visualizations for Energy Compaction

We further visualize the weights $\mathcal{W} = \mathcal{W}_r + j\mathcal{W}_i$ in the frequency temporal learner under different settings, including different lookback window sizes and prediction lengths, on the Traffic and Electricity datasets. The results are illustrated in Figures 7 and 8. These figures demonstrate that

Table 10: Long-term forecasting results comparison with different lookback window lengths $L \in \{96, 192, 336\}$. The prediction lengths are as $\tau \in \{96, 192, 336, 720\}$. The best results are in **bold** and the second best results are underlined.

| Models | FreTS | | PatchTST | | LTSF-Linear | | FEDformer | | Autoformer | | Informer | | Reformer | |
|---|---|---|---|---|---|---|---|---|---|---|---|---|---|---|
| Metrics | MAE | RMSE | MAE | RMSE | MAE | RMSE | MAE | RMSE | MAE | RMSE | MAE | RMSE | MAE | RMSE |
| 96 | **0.037** | **0.051** | 0.039 | 0.052 | 0.038 | 0.052 | 0.050 | 0.067 | 0.050 | 0.066 | 0.066 | 0.084 | 0.126 | 0.146 |
| 192 | **0.050** | **0.067** | 0.055 | 0.074 | 0.053 | 0.069 | 0.064 | 0.082 | 0.063 | 0.083 | 0.068 | 0.088 | 0.147 | 0.169 |
| 336 | **0.062** | **0.082** | 0.071 | 0.093 | 0.064 | 0.085 | 0.080 | 0.105 | 0.075 | 0.101 | 0.093 | 0.127 | 0.157 | 0.189 |
| 720 | **0.088** | **0.110** | 0.132 | 0.166 | 0.092 | 0.116 | 0.151 | 0.183 | 0.150 | 0.181 | 0.117 | 0.170 | 0.166 | 0.201 |
| 96 | **0.036** | **0.050** | 0.037 | 0.051 | 0.038 | 0.051 | 0.067 | 0.086 | 0.066 | 0.085 | 0.109 | 0.131 | 0.123 | 0.143 |
| 192 | **0.051** | **0.068** | 0.052 | 0.070 | 0.053 | 0.070 | 0.080 | 0.101 | 0.080 | 0.102 | 0.144 | 0.172 | 0.139 | 0.161 |
| 336 | **0.066** | **0.087** | 0.072 | 0.097 | 0.073 | 0.096 | 0.093 | 0.122 | 0.099 | 0.129 | 0.141 | 0.177 | 0.155 | 0.181 |
| 720 | **0.088** | **0.110** | 0.099 | 0.128 | 0.098 | 0.122 | 0.190 | 0.222 | 0.191 | 0.224 | 0.173 | 0.210 | 0.159 | 0.193 |
| 96 | **0.038** | **0.052** | 0.039 | 0.053 | 0.040 | 0.055 | 0.088 | 0.113 | 0.088 | 0.110 | 0.137 | 0.169 | 0.128 | 0.148 |
| 192 | **0.053** | **0.070** | 0.055 | 0.071 | 0.055 | 0.072 | 0.103 | 0.133 | 0.104 | 0.133 | 0.161 | 0.195 | 0.138 | 0.159 |
| 336 | **0.071** | **0.092** | 0.074 | 0.099 | 0.077 | 0.100 | 0.123 | 0.155 | 0.127 | 0.159 | 0.156 | 0.193 | 0.156 | 0.179 |
| 720 | **0.082** | **0.108** | 0.100 | 0.129 | 0.087 | 0.110 | 0.210 | 0.242 | 0.211 | 0.244 | 0.173 | 0.210 | 0.168 | 0.205 |

(Row groups correspond to lookback $L = 96$, $L = 192$, $L = 336$ respectively.)

the weight coefficients of the real or imaginary part exhibit energy aggregation characteristics (clear diagonal patterns) which can facilitate frequency-domain MLPs in learning the significant features.

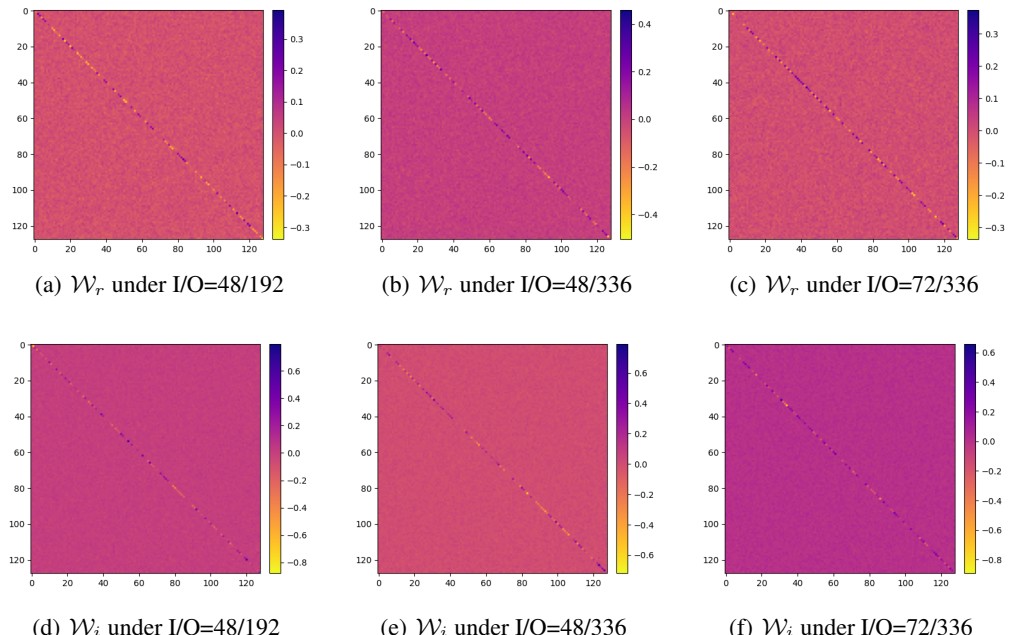

(a) $\mathcal{W}_r$ under I/O=48/192    (b) $\mathcal{W}_r$ under I/O=48/336    (c) $\mathcal{W}_r$ under I/O=72/336

(d) $\mathcal{W}_i$ under I/O=48/192    (e) $\mathcal{W}_i$ under I/O=48/336    (f) $\mathcal{W}_i$ under I/O=72/336

Figure 7: The visualizations of the weights $\mathcal{W}$ in the frequency temporal learner on the Traffic dataset. 'I/O' denotes lookback window sizes/prediction lengths. $\mathcal{W}_r$ and $\mathcal{W}_i$ are the real and imaginary parts of $\mathcal{W}$, respectively.

### G.2 Weight Visualizations for Global View

To verify the characteristics of a global view of learning in the frequency domain, we perform additional experiments on the Traffic and Electricity datasets and compare the weights learned on the input in the time domain with those learned on the input frequency spectrum. The results are presented in Figures 9 and 10. The left side of the figures displays the weights learned on the input in the time domain, while the right side shows those learned on the real part of the input frequency spectrum. From the figures, we can observe that the patterns learned on the input frequency spectrum exhibit more obvious periodic patterns compared to the time domain. This is attributed to the global view characteristics of the frequency domain. Furthermore, we visualize the predictions of FreTS on

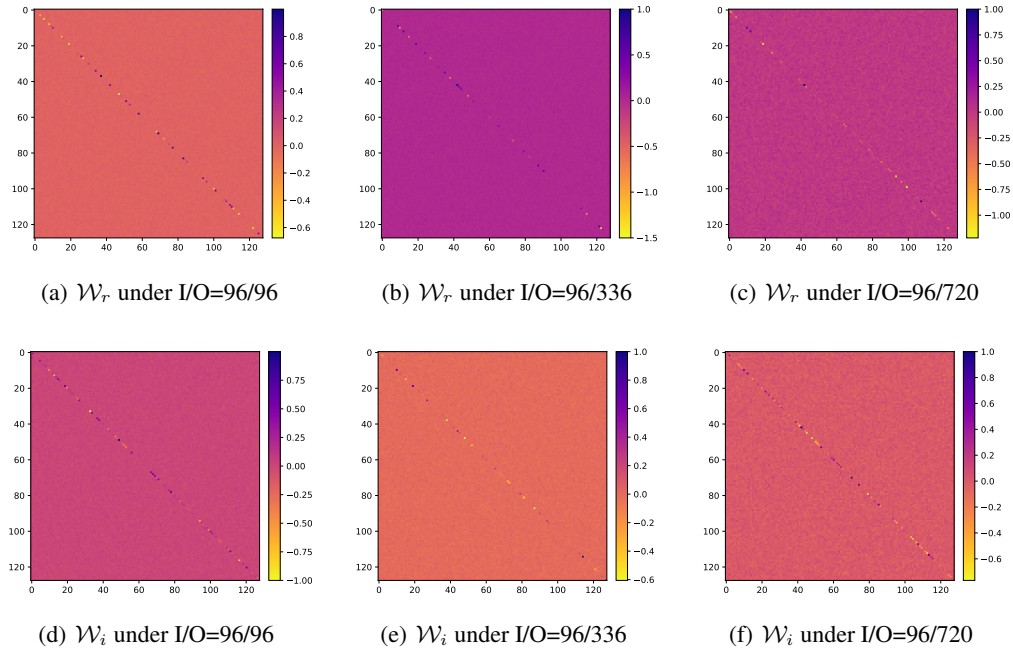

(a) $\mathcal{W}_r$ under I/O=96/96    (b) $\mathcal{W}_r$ under I/O=96/336    (c) $\mathcal{W}_r$ under I/O=96/720

(d) $\mathcal{W}_i$ under I/O=96/96    (e) $\mathcal{W}_i$ under I/O=96/336    (f) $\mathcal{W}_i$ under I/O=96/720

Figure 8: The visualizations of the weights $\mathcal{W}$ in the frequency temporal learner on the Electricity dataset. 'I/O' denotes lookback window sizes/prediction lengths. $\mathcal{W}_r$ and $\mathcal{W}_i$ are the real and imaginary parts of $\mathcal{W}$, respectively.

the Traffic and Electricity datasets, as depicted in Figures 11 and 12, which show that FreTS exhibit a good ability to fit cyclic patterns. In summary, these results demonstrate that FreTS has a strong capability to capture the global periodic patterns, which benefits from the global view characteristics of the frequency domain.

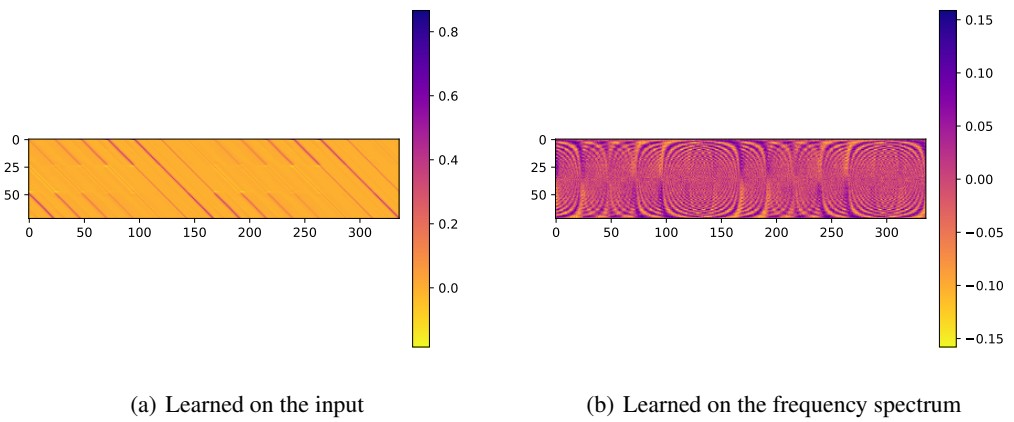

(a) Learned on the input    (b) Learned on the frequency spectrum

Figure 9: Visualization of the weights ($L \times \tau$) on the Traffic dataset with lookback window size of 72 and prediction length of 336.

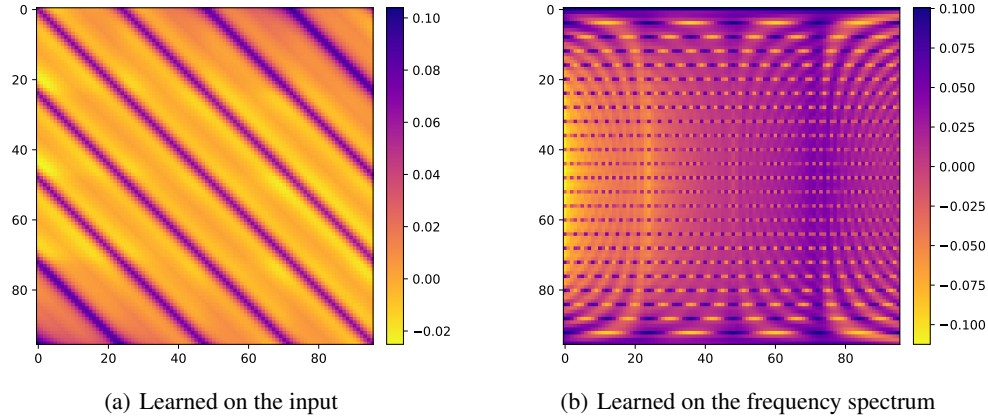

(a) Learned on the input                    (b) Learned on the frequency spectrum

Figure 10: Visualization of the weights ($L \times \tau$) on the Electricity dataset with lookback window size of 96 and prediction length of 96.

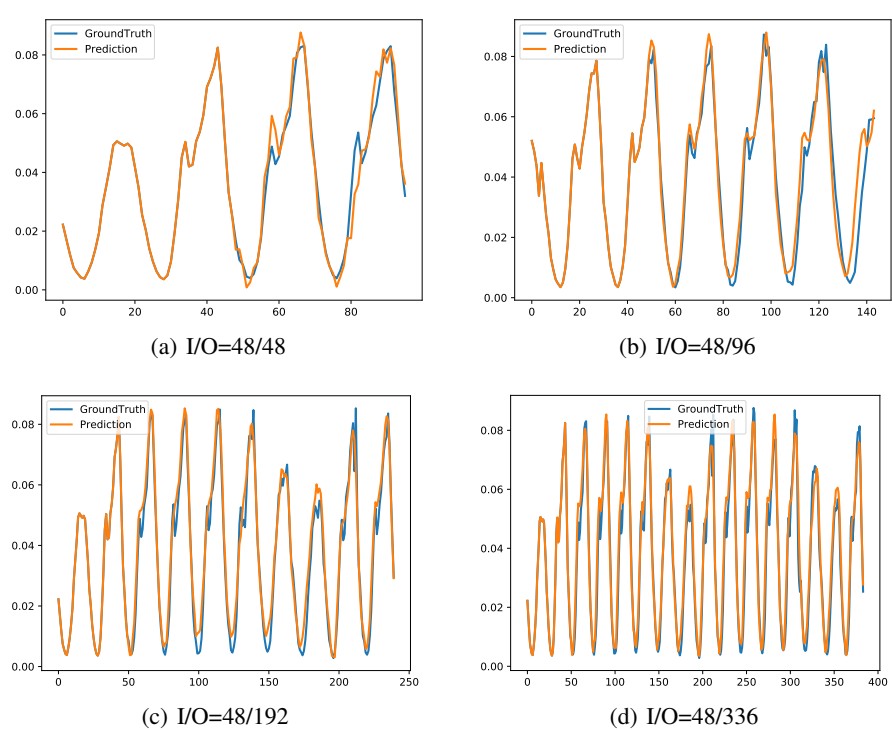

(a) I/O=48/48                    (b) I/O=48/96

(c) I/O=48/192                    (d) I/O=48/336

Figure 11: Visualizations of predictions (forecast vs. actual) on the Traffic dataset. 'I/O' denotes lookback window sizes/prediction lengths.

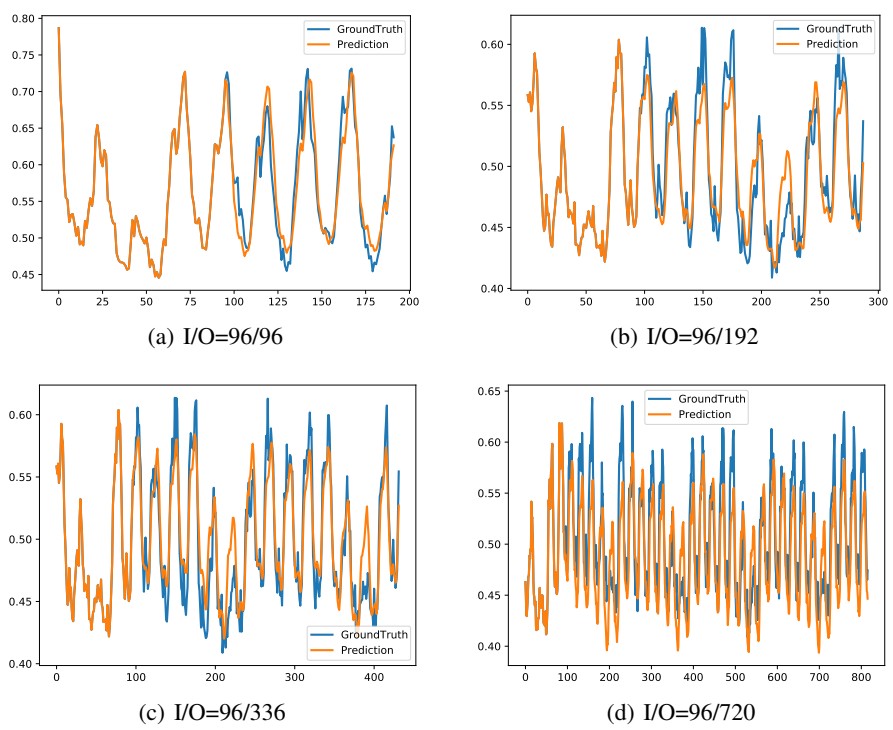

Figure 12: Visualizations of predictions (forecast vs. actual) on the Electricity dataset. 'I/O' denotes lookback window sizes/prediction lengths.

