# OpenReview forum: "Frequency-domain MLPs are More Effective Learners in Time Series Forecasting"
_NeurIPS.cc/2023/Conference — NeurIPS 2023 poster_

### Official Review · Reviewer_bjea · 2023-07-01

**Soundness:** 3 good
**Presentation:** 2 fair
**Contribution:** 2 fair
**Rating:** 5
**Confidence:** 3

**Summary:**

This paper presents FreTS, a framework that addresses both channel-wise and time-wise dependency learning in the frequency domain for time series prediction. FreTS introduces a specially designed frequency-domain MLP structure that processes the real and imaginary parts of the frequency components interactively. The experimental results on 13 benchmarks demonstrate the superior performance of FreTS compared to other state-of-the-art methods.

**Strengths:**

- The two-stage framework and the frequency-domain MLP module introduced in FreTS exhibit some novelty.
- The experimental results across 13 benchmarks show the potential of the proposed method.

**Weaknesses:**

- The authors failed to motivate their method both theoretically and in relation to prior work.  1) The authors did not clearly define and explain the terms "point-wise mappings" and "information bottleneck", not elaborate on how existing methods suffer from these issues, and not clarify how the proposed method addresses and alleviates these problems.  2) While the authors claim that their method has a "global view", it did not highlight how it differs from previous MLP-based and Transformer-based methods in capturing long-range dependencies. 3) The claim of the advantage of “energy compaction” is not supported by experimental evidence.

&nbsp;
- The related work section lacks logical organization and fails to provide a comprehensive summary of previous methods.

&nbsp;
- The method is not well presented. 1) For the Frequency Temporal Learner, the validity behind performing DFT along the channel dimension needs further discussion. It is more likely to be based on engineering considerations than a theoretical grounding. 2) The rationale behind the design of the proposed frequency-domain MLP is unclear regarding why it calculates the new real and imaginary components as Eq. (7).

&nbsp;
- Additional experiments are needed to highlight the benefits of the proposed approach, e.g. the learned global periodic patterns, and the robustness towards noise. Including these experiments will strengthen the empirical evidence for the effectiveness of the proposed method.

&nbsp;
- Certain phrases or sentences lacks clarity, and need improvements. E.g.,  “stacked MLP layers”, “learning in the frequency spectrum helps capture a greater number of periodic patterns”, .etc.

&nbsp;

**Questions:**

- Are the “Domain Conversion” and “Domain Inversion” in Fig. 2 differentiable? How is the network trained end-to-end?
- What are the meanings of the axes in Fig. 1? Whether the results are from a time series or a dataset?


**Limitations:**

The authors are encouraged to discuss the computational complexity after introducing frequency transform and inverse transform into the framework, and its performance on irregular time series.

---

> ### Author Rebuttal · Authors · 2023-08-09
>
> We appreciate your review. Hope our response can address the misunderstandings or concerns
>
> w1.
> 1. The concepts of "point-wise" and "information bottleneck" are widely recognized in the literature. In fact, we provided an explanation of these terms within the context of time series forecasting in **lines 39-42**. The point-wise nature of MLPs limits MLP-based methods' ability to capture long-range global dependencies. Further, these methods suffer from information bottleneck due to the volatile and redundant local momenta of time series, making them hardly capture accurate time-wise dependencies. This work aims to address these issues by leveraging the advantages of frequency-domain MLPs, i.e., global view and energy compaction (see **Lines 43-56**).
> 2. FreTS is based on frequency-domain MLPs, obviously differing from MLP- and Transformer-based methods in architecture for capturing long-range dependencies. Intrinsically, Transformer-based methods rely on pair-wise attention mechanism, MLP-based methods rely on point-wise mapping; while ours rely on the global view of frequency techniques and frequency-domain multiplication.
> 3. On the contrary, "energy compaction" is **discovered** from the experiments. Through experiments (see experimental settings in **Appendix B.4**), we observe that learning in the frequency domain can identify more concentrated diagonal dependencies and patterns than in the time domain (see **Figures 1, 5, 7, and 8**).
>
> W2.
> + FreTS is a frequency-based MLP model for time series forecasting. To provide a comprehensive summary, it is essential to introduce time-domain common models, frequency-based models, and MLP-based models.
> + Correspondingly, the related work is organized from three perspectives: firstly on the time-domain models from classic models to SOTA deep learning models; the second paragraph discusses how existing frequency-based models integrate frequency techniques with neural networks; and the third paragraph describes the representative and SOTA MLP-based models.
>
> This way logically organizes the models relevant to our work with a comprehensive discussion of the previous related literature.
>
> W3.
> 1. Note that Frequency Temporal Learner performs DFT on the time dimension (**see line 165**) while Frequency Channel Learner is on the channel dimension (**see line 152**). We guess you may ask about the theoretical grounding for Frequency Channel Learner, please refer to the general response about the frequency channel learner.
> 2. In the frequency domain, the values representing the frequency spectrum are complex numbers. Mathematically, the multiplications in the frequency domain adhere to **the basic rule of complex multiplication**. Please refer to **Appendix C** where we show a calculation example.
>
> W4.
> 1. Note that we have already conducted extensive experiments to verify our claims regarding the benefits of the proposed approach.
> + To verify the global view of learning in the frequency domain, we performed visualization experiments on the Traffic and Electricity datasets, comparing the weights learned in the time domain with those learned on the frequency spectrum.
> The results can be shown in **Figures 9, 10, 11, and 12 in Appendix G.2**, verifying that FreTS has a strong capability to capture the global periodic patterns.
> + To verify energy compaction in the frequency domain, we visualized the weights in the frequency temporal learner on Traffic and Electricity. The results can be reported in **Figures 5, 7, and 8**, exhibiting energy aggregation characteristics with clear diagonal patterns and dependencies.
>
> 2. To address your concern regarding the robustness of FreTS towards noise, we add $0.1 \times \mathcal{N}(0,1)$ Gaussian random noise into training data on the ETTh1 dataset. The results shown in the following table support our claim of robustness.
>
> | |without noise | |with noise | |
> |:--|:--|:--|:--|:--|
> | Metric| MAE| RMSE| MAE| RMSE|
> |96|0.061|0.087|0.061|0.087|
> |192|0.065|0.091|0.065|0.091|
> |336|0.070|0.096|0.070|0.097|
> |720|0.082|0.108|0.083|0.109|
>
> W5.
> In **Line 94**: "stacked MLP layers" refers to N-BEATS proposes a deep architecture based on backward and forward residual links and a very deep stack of fully-connected layers (**cf. line 2-3 in Abstract of N-BEATS paper**).
> In **Line 129-130**: we state "learning in the frequency spectrum helps capture a greater number of periodic patterns" to explain the observation from Figure 1(a), meaning the patterns learned in the frequency domain exhibit more obvious global periodic patterns than in the time domain. We have explained this in **the caption of Figure 1** and in **lines 50-51**. We will carefully polish the phrases.
>
> Q1.
> Certainly, both of them are differentiable, please refer to **Equations (1) and (2)**. Our FreTS is an end-to-end forecasting model.
>
> Q2.
> Fig. 1 visualizes the learned weights $W \in \mathbb{R}^{d\times d}$ where $d$ is the hidden dimension. The axes are the dimension indices of the weight matrix.
> The weights for visualizing Fig. 1 are learned on the Traffic dataset (see **lines 575 and 584 in Appendix**). A detailed description of the visualization settings is **in Appendix B.4**.
>
> Limitation:
>
> 1. In **Section 4.3**, we have conducted experiments to evaluate the efficiency of FreTS, covering short-term forecasting scenarios and long-term forecasting scenarios, in comparison with SOTA models. Contrary to your concern, incorporating Fourier transform can improve efficiency (see reasons in our responses to Reviewer WVuT). Also, this advantage has been widely explored and acknowledged in the literature for time series forecasting, such as Autoformer and FEDformer.
>
> 2. In this paper, similar to all baseline methods, we only focus on time series forecasting without focusing on irregular time series. Frequency transformation may also help address irregular time series issues, and we leave this as our future work.
>
> We'll clarify the above in the final version.

---

> > ### Comment · Reviewer_bjea · 2023-08-13
> >
> > I appreciate the authors' efforts to clarify the concerns with more detailed explanations. My questions are mostly addressed, and I have raised my score.

---

> > > ### Author Response · Authors · 2023-08-13
> > > **Thanks for feedback**
> > >
> > > Dear Reviewer bjea, we are appreciated to receive your feedback. Many Thanks.
> > >
> > > Authors

---

### Official Review · Reviewer_u93d · 2023-07-03

**Soundness:** 3 good
**Presentation:** 3 good
**Contribution:** 3 good
**Rating:** 6
**Confidence:** 4

**Summary:**

This paper investigates time series forecasting in the frequency domain. By utilizing MLPs in the frequency domain, the proposed FreTS effectively captures the patterns of time series with a global view and energy compaction. Frequency learning is applied to both inter-series and intra-series scales, allowing FreTS to capture channel-wise and time-wise dependencies. Extensive empirical experiments demonstrate the effectiveness of FreTS in both short-term and long-term forecasting tasks.


**Strengths:**

1) This paper redesigns MLPs in the frequency domain to effectively capture both time-wise and channel wise correlations. The use of simple MLPs ensures high efficiency and helps mitigate overfiting issues compared to existing sophisticated deep models.
2) This paper is well-organized, with comprehensive discussion and experiments.

**Weaknesses:**

1) The authors mention that learning time series in the frequency domain has the nature of energy compaction, as the energy concentrates on the smaller part of frequency components. However, the proposed FreTS still retains all the frequency components when performing MLP in the frequency domain, thereby not fully leveraging this advantageous characteristic.
2) Frequency domain modeling is advantageous for capturing temporal dependencies due to the inherent periodic patterns in time series. However, the suitability of frequency domain modeling for exploring channel-wise correlations requires further discussion and verification. Moreover, conducting channel-wise modeling for each time point is inefficient, and should be performed at a coarser granularity, such as segment-wise or series-wise.
3) The dimension extension block takes an input $X_t \in \mathbb{R}^{N \times L}$ and produces an output $H_t \in \mathbb{R}^{N \times L \times d}$. However, if the number of variables $N$ or the input length $L$ is large, the memory usage of $H_t$ can become very high.

**Questions:**

1) Short-term forecasting comparison shown in Table 1 is suggested to include N-BEATS[1] and N-HITS[2] as baselines as well.
2) Figure 4(b) demonstrates that the parameter number of FreTS remains constant as the prediction length increases. However, the projection block described in subsection 3.1 includes learnable parameters $\phi_2 \in \mathbb{R}^{d_h \times \tau}$ and $b_2 \in \mathbb{R}^\tau$, and the number of these parameters will increase with the prediction length $\tau$.
3) From which learner (frequency channel or frequency temporal), and at which layer are the weights used for obtaining the visualizations?

Reference

[1] Oreshkin B N, Carpov D, Chapados N, et al. N-BEATS: Neural basis expansion analysis for interpretable time series forecasting[C]//International Conference on Learning Representations. 2019.
[2] Challu C, Olivares K G, Oreshkin B N, et al. NHITS: Neural Hierarchical Interpolation for Time Series Forecasting[C]//Proceedings of the AAAI Conference on Artificial Intelligence. 2023, 37(6): 6989-6997.

---

> ### Author Rebuttal · Authors · 2023-08-09
>
> We appreciate your review and the positive comments regarding our paper. Below, we address your comments.
>
> **W1. ...still retains all the frequency components..., thereby not fully leveraging this advantageous characteristic.**
>
> We preserve the entire frequency components and feed them into the frequency channel/temporal learners to avoid any information loss, which is necessary and rational. Note that all components, such as high-frequency ones or low-frequency ones, are adaptively highlighted or downweighed according to different data characteristics.
> Accordingly, *energy compaction* is achieved by adaptively learning in the frequency domain instead of manually discarding some components; it means that the information is concentrated on a smaller portion of components than in the time domain. This may be attributed to the fact that the frequency spectrum has discriminative frequency components specified for different data characteristics.
>
> **W2**
> 1. Frequency domain modeling is theoretically and empirically suitable for exploring channel-wise correlations.
>   + Theoretically, the frequency channel learner is equivalent to applying global convolutions on the variables for each timestamp (see **Appendix D.2 Proof of Theorem 2**).
>   + Empirically, we have supplemented visualization experiments to verify the channel-wise learning capability of our frequency channel learner. For more details, **please refer to the general response about Frequency channel learner**.
>
> 2. In many real-world scenarios, such as traffic scenarios, channel-wise dependencies can be time-evolving, even varying at different time points. In such cases, it becomes necessary and beneficial to conduct channel-wise modeling for each time point individually to capture the time-varying channel-wise dependencies. This approach is commonly employed in the literature, like in AGCRN and StemGNN.
>
>     Modeling channel-wise dependencies at a segment-wise or series-wise granularity is a promising idea, reminiscent of the successful implementation of series-wise time-wise dependencies in Autoformer. However, this approach overlooks the temporal dynamics within the segment and should integrate time information for comprehensive modeling. Due to time constraints, we were unable to conduct experiments to validate this idea. Nevertheless, we appreciate your constructive suggestion and will take it into account for our future research endeavors.
>
> **W3. ...if the number of variables or the input length is large, the memory usage can become very high.**
>
> First, performing time series embedding is widely adopted in recent MTS forecasting models, e.g., StemGNN and AGCRN. Second, we deliberately choose a smaller hidden dimension, specifically $d=128$, in comparison to Transform-based models that typically employ higher dimensions, such as 1024. As a result, the memory usage of $\mathbf{H}_t$ is perfectly acceptable, which is evidenced by the fact that our FreTS capably works on long-term forecasting on all datasets and achieves the SOTA performance.
>
> In addition, to avoid **extensive parameters** in the dimension extension block, we employ $\mathbf{H}_t=\mathbf{X}_t \times \phi_d$ where $\phi_d \in \mathbb{R}^{1 \times d}$ (see **lines 124-125**). Note that the parameter volume of the embedding matrix of $\phi_d$ is independent of $N$ and $L$. This approach effectively avoids extensive parameters, as demonstrated in Figure 4(a), where the parameter count of FreTS remains unaffected by changes in $N$.
>
> **Q1. Short-term ... comparison ... suggested to include N-BEATS[1] and N-HITS[2] as baselines...**
>
> Thank you for pointing out the two models. In Related Work, we have discussed these models, but we did not compare them with FreTS, considering they are univariate forecasting models. Taking your suggestion, we conducted a comparison between FreTS and the two baselines under the input length and prediction length of 12. The corresponding results are presented below:
>
> | | Solar| |Wiki | |Traffic| |ECG| |Electricity| |COVID-19| |METR-LA| |
> |:---|:---|:---|:---|:---|:---|:---|:---|:---|:---|:---|:---|:---|:---|:---|
> | | MAE| RMSE|MAE | RMSE|MAE| RMSE|MAE| RMSE|MAE| RMSE|MAE|RMSE |MAE| RMSE|
> |N-BEATS|0.137|0.182|0.047|0.087|0.016|0.031|0.056|0.088|0.057|0.085|0.162|0.192|0.087|0.166|
> |N-HITS|0.129 |0.174 |0.046|0.083|0.018|0.035|0.055|0.086|0.060|0.089|0.155|0.183|0.090|0.173
> |FreTS|0.120|0.162|0.041|0.074|0.011|0.023|0.053|0.078|0.050|0.076|0.123|0.167|0.080|0.166|
>
> The results show that FreTS surpasses N-BEATS and N-HITS across all datasets, thereby demonstrating that frequency-domain MLP is more effective than time-domain MLP. We will include these results in the final version.
>
> **Q2. ...the number of these parameters will increase with the prediction length**
>
> Thank you for reading our paper carefully. Indeed, the number of parameters in FreTS increases with the prediction length $\tau$. The raw number of parameters in FreTS for drawing Figure 4(b) is 3.23, 3.26, 3.30, 3.34. However, due to the minimal variation among these data points, the curve exhibits only slight changes.
>
> We will redraw the figure and put values on the figure to show the change more clearly.
>
> **Q3. From which learner ... which layer are the weights used for obtaining the visualizations?**
>
> Note that both the frequency channel learner and temporal learner adopt one layer of FreMLP (see **line 556 in Appendix B.3**), the reason for one layer is explained in **our response to the question W1 by Reviewer aNLj**. To ensure a fair comparison with DLinear, we follow the visualization settings used in DLinear, where we utilized the weights of frequency temporal learner for visualizations, the same as DLinear does. Further details about the visualization settings are in **Appendix B.4**, and we have uploaded the source codes of the visualization method in the supplementary materials for reference.
>
> We'll clarify the above in the final version.

---

> > ### Comment · Reviewer_u93d · 2023-08-14
> >
> > Thanks for the detailed responses to the initial reviewing comments. Generally, the authors have addressed most of my concerns.
> > Nevertheless, in the response to W1, it is mentioned that the model can adaptively highlight or downweigh different frequency components. Maybe further clarification is still expected on this point. Generally MLPs are generally shared across different frequency components, but FreMLP only consists of one single layer.

---

> > > ### Author Response · Authors · 2023-08-14
> > > **Thanks for your feedback**
> > >
> > > Dear Reviewer u93d,
> > >
> > > We are appreciated to receive your feedback. We would like to clarify the point you mentioned.
> > >
> > > - Note that in FreMLPs, the weights $\mathcal{W}$ are complex numbers that consist of two parts of weights, i.e., $\mathcal{W}=\mathcal{W}_r+j\mathcal{W}_i$, where $\mathcal{W}_r \in \mathbb{R}^{d\times d}$ is the real part and $\mathcal{W}_i \in \mathbb{R}^{d\times d}$ is the imaginary part (see **Definition 1 and Equation (7)**). Correspondingly, the multiplication in the frequency domain is implemented by the separate computation on the real and imaginary parts of frequency components, adhering to the **basic rule of complex multiplication** (see **Equation (7) and Appendix C**).
> > >
> > > - According to the convolution theorem [1], the Fourier transform of a convolution between two sequences is equal to the pointwise multiplication of their respective Fourier transforms. This theorem enables us to efficiently conduct convolutions in the frequency domain. In mathematics, the calculation in FreMLP in the frequency domain, i.e., $\mathcal{H}\mathcal{W}$ where the input $\mathcal{H}$ and the weights $\mathcal{W}$ are complex-valued, involves applying a filter of $\mathcal{W}$ over $\mathcal{H}$, which is equivalent to performing a convolution in the time domain (see **Theorem 2**).
> > >
> > > - Based on the above, the calculations in FreMLPs exhibit distinctions when compared to general MLPs in the time domain. General MLPs can be regarded as performing transformations using the MLP weights to highlight or downweigh certain features (elements), **while FreMLPs can be regarded as applying filtering in the frequency domain over the frequency spectrum** to highlight or downweigh certain frequency components.
> > >
> > > - A tiny illustration example, for a real-valued vector $\mathbf{V}$, multiplying a real number $r$ with $\mathbf{V}$, e.g., $2\mathbf{V}$ with $r=2$, means equally highlighting/downweighing all elements in the vector by $r$; while for a **complex-valued** vector $\mathcal{V}$, multiplying a **complex number** $c$ with $\mathcal{V}$, e.g., $(2+2j)\mathcal{V}$ with $c=(2+2j)$, does not mean equally highlighting/downweighing all elements in the vector. Instead, the multiplication can be **regarded as performing the filter of $c$ onto the frequency spectrum $\mathcal{V}$**.
> > >
> > > In summary, although we adopt shared complex number weight matrices in each layer of FreMLP, FreMLP can adaptively learn to highlight some frequency components and downweigh different frequency components during training.
> > >
> > > [1]. S. S. Soliman and MD Srinath. Continuous and discrete signals and systems. Prentice Hall, (1990)
> > >
> > > - **Reasons for the one-layer setting in FreMLP**: a one-layer FreMLP is powerful enough and empirically performs well to learn channel-wise or time-wise dependencies while stacking multiple FreMLP brings more parameters and may lead to lower learning efficiency.
> > > For more experimental details, please refer to our responses to **W2** and **Q2** of Reviewer aNLj.
> > >
> > > Hope we have addressed your concerns. If you have any further questions or concerns, please feel free to let us know.
> > >
> > > Authors

---

> > > > ### Comment · Reviewer_u93d · 2023-08-17
> > > >
> > > > Thank you for the further explanation. I would be happy to raise my score accordingly.

---

> > > > > ### Author Response · Authors · 2023-08-17
> > > > >
> > > > > Dear Reviewer u93d, we greatly appreciate your continued valuable feedback. If you have any further questions or concerns, please feel free to get in touch with us.
> > > > >
> > > > > Authors

---

### Official Review · Reviewer_BMVc · 2023-07-04

**Soundness:** 4 excellent
**Presentation:** 4 excellent
**Contribution:** 3 good
**Rating:** 6
**Confidence:** 4

**Summary:**

In this paper, the authors investigate the problem of time-series forecasting. Since the frequency domain can preserve the information from a global view and enjoy the advantage of energy compaction, the authors propose the FreTS model, which is composed of the Frequency Channel Learner and the Frequency Temporal Learner. The authors further prove the reasonableness of the frequency-domain MLPs and evaluate the proposed idea on several datasets.

**Strengths:**

1.	The authors investigate the time-series forecasting problem from the frequency domain and provide convincing reasons for the advantage of the frequency domain.
2.	The authors devise the frequency channel learner and frequency temporal learner, which address different challenges in a unified framework.
3.	The authors investigate the proposed method on several datasets and achieve the ideal performance.


**Weaknesses:**

1.	To capture the dependencies among channels, the authors propose the frequency channel learner, which applies the FreMLP on the variables from each timestamp, e.g. $H_t^:$. However, Since it is strange to consider $H_t^:$ as a time series, I think it might be not a good idea to employ FreMLP to capture these dependencies. In my opinion, I think the CNNs, the attention mechanism, and GNN might be good choices. It is suggested that the author should try different methods to better capture these dependencies.
2.	The authors address the frequency-domain MLP from time-series forecasting, but some time-series data might be not seasonal and contain some monotonic tendency, can the proposed method address this problem?


**Questions:**

Please refer weaknesses

**Limitations:**

Please refer weaknesses

---

> ### Author Rebuttal · Authors · 2023-08-09
>
> We appreciate your review and the positive comments on our work. We address each of them as follows.
>
> **W1. To capture the dependencies among channels, we propose the frequency channel learner, which applies the FreMLP on the variables from each timestamp, e.g., $\mathbf{H}_t^{:}$. However, since it is strange to consider $\mathbf{H}_t^{:}$ as a time series, I think it might be not a good idea to employ FreMLP to capture these dependencies. In my opinion, I think the CNNs, the attention mechanism, and GNN might be good choices. It is suggested that the author should try different methods to better capture these dependencies.**
>
> A1. Theoretically, the frequency channel learner is equivalent to applying global convolutions on the variables for each timestamp. Empirically, we have performed visualization experiments to verify the channel-wise learning capability of our frequency channel learner.
> **Please refer to general response about Frequency channel learner**.
>
> **W2. The authors address the frequency-domain MLP from time-series forecasting, but some time-series data might be not seasonal and contain some monotonic tendency, can the proposed method address this problem?**
>
> A2. Yes, FreTS can learn the monotonic tendency, we have analyzed the reasons and conducted experiments to verify this. **Please refer to the general response about FreTS on non-periodic data**.
>
>
> We'll clarify the above in the final version and hope that we have addressed all your concerns. Thanks.

---

> ### Comment · Area_Chair_bPc6 · 2023-08-19
> **Gentle Reminder by AC**
>
> Dear Reviewer,
>
> Could you carefully read the authors' rebuttal as well as the others' reviews and their rebuttal, and make responses at your earliest convenience? The deadline for the discussion phase is fast approaching, which is due Aug 21st 1pm EDT, so your quick responses will be greatly appreciated.
>
> Best,
>
> AC

---

### Official Review · Reviewer_aNLj · 2023-07-06

**Soundness:** 3 good
**Presentation:** 3 good
**Contribution:** 3 good
**Rating:** 7
**Confidence:** 4

**Summary:**

The authors argue that MLP-based forecasting methods suffer from point-wise mappings and information bottlenecks and explore an interesting direction of applying MLPs in the frequency domain for time series forecasting. They further analyze the inherent characteristics of frequency-domain MLPs and propose the FreTS model via stacking frequency-domain MLPs for time series forecasting. The paper provides theoretical guarantees and extensive empirical evaluation to analyze the advantages of frequency-domain MLPs and show the superiority of FreTS, verifying the authors’ arguments and the effectiveness of frequency-domain MLPs.

**Strengths:**

1. The authors study an interesting neural network of frequency-domain MLPs and analyze the advantaged effectiveness of frequency-domain MLPs compared MLPs in the time domain. This potentially inspires the MTS forecasting community to pay more attention to frequency analysis.
2. Frequency-domain MLPs are theoretically proved to be energy compacting and equivalent to global convolutions. FreTS based on frequency-domain MLPs is straightforward in architecture and theoretically sound.
3. The experimental results are comprehensive and impressive. Extensive results on 13 datasets validate the superiority of FreTS and the effectiveness of frequency-domain MLPs for both short-term forecasting and long-term forecasting. Model analysis and efficiency analysis are also provided to investigate the advantages of FreMLP over MLP and the higher efficiency of FreTS over SOTA baselines.
4. The visualization analysis provides quite interesting patterns and clearly shows the distinct characteristics between the time domain (MLPs) and the frequency domain (frequency-domain MLPs). Especially, the visualizations of the learned weights in the frequency domain show highly concentrated values in the diagonal, verifying the energy compacting and learning efficiency of frequency-domain MLPs.
5. The paper is well-organized and easy to follow, and the main contributions are quite clear and solid. In my opinion, it studies an interesting topic and provides a new perspective on incorporating frequency analysis into time series analysis.

**Weaknesses:**

1. The experimental results are convincing, but only one MLP-based baseline is compared in the experiments. Additional MLP-based baselines help to verify the advantages of FreMLPs over MLPs.
2. The superiority performance of FreTS verifies the effectiveness of (one layer) frequency-domain MLPs, but the authors did not investigate how well multi-layers of frequency-domain MLPs perform in MTS forecasting.

**Questions:**

1. The learned patterns on the frequency spectrum show that FreTS can capture more obvious periodic patterns. But when there are no periodic patterns in MTS data, can FreTS still perform well? That is, whether FreTS is suitable for non-periodic MTS data.
2. Why FreTS contains only one layer of FreMLP in either frequency channel learner and frequency temporal learner? How well does FreTS work when stacking with multiple FreMLPs?

===

I have read the rebuttal and would like to keep my rating.

---

> ### Author Rebuttal · Authors · 2023-08-09
>
> Many thanks for your constructive comments and suggestions. We provide a point-by-point response to your comments below.
>
> **W1. The experimental results are convincing, but only one MLP-based baseline is compared in the experiments. Additional MLP-based baselines help to verify the advantages of FreMLPs over MLPs.**
>
> A1. Thank you for your suggestion. We have incorporated additional experiments to compare FreTS with two state-of-the-art MLP-based baselines, namely N-BEATS [1] and N-HITS [2], in the context of short-term forecasting. The results of these comparisons are provided below.
>
> | | Solar| |Wiki | |Traffic| |ECG| |Electricity| |COVID-19| |METR-LA| |
> |:---|:---|:---|:---|:---|:---|:---|:---|:---|:---|:---|:---|:---|:---|:---|
> | | MAE| RMSE|MAE | RMSE|MAE| RMSE|MAE| RMSE|MAE| RMSE|MAE|RMSE |MAE| RMSE|
> |N-BEATS|0.137|0.182|0.047|0.087|0.016|0.031|0.056|0.088|0.057|0.085|0.162|0.192|0.087|0.166|
> |N-HITS|0.129 |0.174 |0.046|0.083|0.018|0.035|0.055|0.086|0.060|0.089|0.155|0.183|0.090|0.173
> |FreTS|0.120|0.162|0.041|0.074|0.011|0.023|0.053|0.078|0.050|0.076|0.123|0.167|0.080|0.166|
>
> The results show that FreTS surpasses N-BEATS and N-HITS across all datasets, thereby demonstrating that frequency-domain MLP is more effective than time-domain MLP. We will include these results in the final version.
>
> Additionally, we have conducted a comparison between FreTS and another MLP-based baseline LightTS [3] on long-term forecasting settings. The results of this comparison are presented in the table below:
>
> | Dataset  | |ETTh1 | |  | | ETTm1| |  | |
> |:---------|:---|:---|:---|:---|:---|:---|:---|:---|:---|
> | Models  |Metric |96| 192| 336 | 720|96| 192| 336 | 720|
> | LightTS|MAE|0.063 | 0.066| 0.072| 0.085| 0.058| 0.061|0.065 | 0.073|
> | |RMSE | 0.090|0.094 |0.099 | 0.112| 0.084| 0.088| 0.092| 0.101|
> | FreTS|MAE| 0.061| 0.065| 0.070| 0.082| 0.052| 0.057| 0.062| 0.069|
> | | RMSE| 0.087| 0.091| 0.096| 0.108| 0.077| 0.083| 0.089| 0.096|
>
> The results obtained from both the short-term forecasting and long-term forecasting experiments clearly demonstrate that FreTS outperforms the time domain MLP-based baselines. This indicates the superiority of the proposed frequency-domain MLPs over their time-domain counterparts in terms of forecasting performance.
>
> We will add the above MLP-based baselines in the experiment and included the results in the final version.
>
> [1] N-BEATS: Neural basis expansion analysis for interpretable time series forecasting, ICLR. 2019.
> [2] NHITS: Neural Hierarchical Interpolation for Time Series Forecasting, AAAI. 2023.
> [3] Less Is More: Fast Multivariate Time Series Forecasting with Light Sampling-oriented MLP Structures. arXiv, 2022.
>
> **W2. The superiority performance of FreTS verifies the effectiveness of (one layer) frequency-domain MLPs, but the authors did not investigate how well multi-layers of frequency-domain MLPs perform in MTS forecasting.**
>
> A2. Theoretically, FreMLP shows an inherent characteristic of energy compaction, as shown in **Theorem 1** . It means FreMLP can effectively extract the key patterns while reducing other redundant information, leading to improved generalization and better performance. Intuitively, a one-layer FreMLP is powerful enough and performs well to learn channel-wise or time-wise dependencies, while stacking multiple FreMLP may brings more parameters which reducing learning efficiency. To verify this, we conduct additional experiments on Exchange dataset under the input length of 96 and the prediction length of 96. The results are shown in the following table:
>
> | Metric  | one layer | two layers| three layers | four layers|
> |:--------|:---------|:---|:---|:---|
> | MAE  |  0.037       |  0.037| 0.037|0.038|
> | RMSE |   0.051      | 0.051|0.051|0.052|
>
> From the results, we can observe that FreTS achieves the same performance when utilizing \{1,2,3\} layer temporal learner. The results validate our claim that a one-layer FreMLP is adequate for constructing the channel learner and temporal learner to enable the acquisition of channel-wise and time-wise dependencies, respectively.
>
> We will supplement the experiments of hyperparameter analysis in the appendix of our final version.
>
> **Q1. The learned patterns on the frequency spectrum show that FreTS can capture more obvious periodic patterns. But when there are no periodic patterns in MTS data, can FreTS still perform well? That is, whether FreTS is suitable for non-periodic MTS data.**
>
> A3. **Please refer to general response about FreTS on non-periodic data**.
>
> **Q2. Why FreTS contains only one layer of FreMLP in either frequency channel learner and frequency temporal learner? How well does FreTS work when stacking with multiple FreMLPs?**
>
> A4. Note that during our experiments,  we conducted careful hyperparameter tuning for FreTS. Specifically, we focused on tuning the number of layers in both the frequency channel learner and the frequency temporal learner. we found that using a one-layer channel learner and a one-layer temporal learner yielded the best forecasting performance for FreTS.
>
> In addition, we have provided the corresponding results of FreTS for different numbers of layers, specifically {1, 2, 3, 4}, in the above table in the response to **W2**. From the results, we can observe that FreTS achieves the best performance when adopting one-layer learner.
>
> We'll update the paper to address the above aspects and hope we have addressed your comments.

---

> > ### Comment · Reviewer_aNLj · 2023-08-16
> > **Reply to rebuttal**
> >
> > Thanks for the authors' effort in providing the rebuttal. This helped clarify my concerns. I am happy to keep my current rating.

---

> > > ### Author Response · Authors · 2023-08-16
> > > **Thanks for your feedback**
> > >
> > > Dear Reviewer aNLj, we are appreciated to receive your feedback. If you have any further questions or concerns, please feel free to let us know.
> > >
> > > Authors

---

### Official Review · Reviewer_WVuT · 2023-07-22

**Soundness:** 3 good
**Presentation:** 3 good
**Contribution:** 3 good
**Rating:** 6
**Confidence:** 5

**Summary:**

The paper studies time series forecasting problem under the deep learning paradigm. Authors propose a new network architecture with MLPs in the frequency domain to capture both inter-series and intra-series correlations.

**Strengths:**

1.	The paper is well-written and easy to follow.
2.	Learning the spatio-temporal correlations in the frequency with MLPs seems reasonable and shows lightweight in the experiment part.
3.	Experiments with diverse datasets and solid comparison methods demonstrate the effectiveness of the proposed method.


**Weaknesses:**

1.	As mentioned in the related work part, there are already several works in the frequency domain for time series forecasting. It should be clearly discussed what’s the difference of this work.
2.	Since the efficiency is a core merit of the proposed method, it would be better to analyze the theoretical computation complexity compared with other method.
3.	While the frequency method may improve the efficiency, it is not clear whether these kind of methods would mainly focus on the low-frequency part but ignoring the high-frequency part, which is more important for time series forecasting.


**Questions:**

Please check the weakness

---

> ### Author Rebuttal · Authors · 2023-08-09
>
> We appreciate your review and the positive comments regarding our paper. We would like to respond to your comments as follows.
>
> **W1. As mentioned in the related work part, there are already several works in the frequency domain for time series forecasting. It should be clearly discussed what’s the difference of this work.**
>
> A1. Thank you for your suggestion. In the Related Work section, we have presented various existing frequency-based models that incorporate frequency techniques into neural networks. These models include SFM, which combines DFT with LSTM; StemGNN, which incorporates DFT with GNN; and Autoformer and FEDformer, which use DFT with self-attention in the Transformer architecture. These models are considered **frequency-enhanced architectures** as they leverage frequency techniques to improve upon the original architecture, such as Transformer. However, our work differs from these approaches since we propose a **frequency learning architecture** that learns channel-wise and time-wise dependencies in the frequency domain through a novel frequency-domain MLP network.
>
> To clearly distinguish our work from the previous frequency-based models, we will clarify the comprehensive discussion highlighting the key differences and distinctions in the Related Work section.
>
>
> **W2. Since the efficiency is a core merit of the proposed method, it would be better to analyze the theoretical computation complexity compared with other method.**
>
> A2. In the following table, we analyze the theoretical complexity compared with other representative SOTA models:
>
> |Long-term| |Short-term| |
> |:---|:---|:---|:---|
> |Model|Complexity|Model|Complexity|
> |FEDformer|$\mathcal{O}(L)$ | AGCRN| $\mathcal{O}(N^2)$|
> |Autoformer|$\mathcal{O}(L\operatorname{log}L)$|MTGNN|$\mathcal{O}(N^2)$|
> |PatchTST| $\mathcal{O}((L/P)^2)$| StemGNN| $\mathcal{O}(N^3)$|
> |||FreTS| $\mathcal{O}(N\operatorname{log}N+L\operatorname{log}L)$ |
>
> From the table, we can see that our complexity is log-linear. We will add the theoretical computation complexity to the final version.
>
> **W3. While the frequency method may improve the efficiency, it is not clear whether these kind of methods would mainly focus on the low-frequency part but ignoring the high-frequency part, which is more important for time series forecasting.**
>
> A3. Note that, **after frequency transformation, we do not discard any frequency components nor specifically focus on the low-frequency components but ignoring the high-frequency components**. Instead, we preserve the entire frequency components and feed them into the frequency channel/temporal learners. Note that all components, including high-frequency ones or low-frequency ones, are adaptively highlighted or downweighed specific to different data characteristics.
>
> 1. The core operation of our FreTS is the frequency-domain MLP (FreMLP) that performs multiplications in the frequency domain and is equal to global convolutions in the time domain (refer to **Appendix D**). The efficiency of our FreMLP arises from two primary factors:
> - First, frequency transformation often leads to a sparse frequency spectrum specific to data characteristics, resulting in a significant portion of the frequency components, including both low-frequency and high-frequency ones, approaching zero. This enables FreTS to adaptively ignore negligible components and highlight significant components based on the characteristics of the data.
> - Second, according to the convolution theorem [1], the Fourier transform of a convolution between two sequences is equal to the pointwise multiplication of their respective Fourier transforms. This theorem enables us to efficiently conduct convolutions in the frequency domain, leading to improved computational efficiency.
>
> In summary, both the sparse frequency spectrum and the point-wise product in the frequency domain contribute to the efficient FreTS (FreMLP) model.
>
> 2. In addition to efficiency, our FreTS model also benefits from frequency techniques to improve forecasting effectiveness.
> - First, the calculation of a frequency spectrum involves summing all signals across time, resulting in each spectrum element in the frequency domain attending to all timestamps in the time domain. This characteristic indicates that a spectrum provides a global view of the entire sequence of time series, which is advantageous for capturing global patterns, such as global periodic patterns, which are crucial for effective time series forecasting. This is demonstrated in **Figures 1(a), 9, 10, 11, and 12**.
> - Furthermore, frequency transformation also exhibits a characteristic of *energy compaction*, whereby the essential features of signals are typically represented by a subset of frequency coefficients that have significantly smaller magnitudes compared to the original signal. This characteristic helps reduce data redundancy and facilitates identifying more important features and clearer patterns (as shown in **Figures 1(b), 5, 7, and 8**).
>
> Reference:
>
> [1]. S. S. Soliman and MD Srinath. Continuous and discrete signals and systems. Prentice Hall, (1990)
>
> We'll clarify the above in the final version and hope we have addressed your comments.

---

> > ### Comment · Reviewer_WVuT · 2023-08-16
> >
> > Thanks for the author's responses, which have solved my main concerns. Thus, I would like to raise the score to weak accept.

---

> > > ### Author Response · Authors · 2023-08-17
> > > **Thanks for your feedback**
> > >
> > > Dear Reviewer WVuT, we sincerely value your feedback and the constructive suggestions you've provided for enhancing our paper. If you have any further questions or concerns, please feel free to let us know.
> > >
> > > Authors

---

### Author Rebuttal · Authors · 2023-08-09

Dear Reviewers, ACs and the SAC:

We thank you all for the review and valuable comments. We'll clarify them in the final version to address all relevant questions and constructive suggestions.

To address the common concerns regarding our frequency channel learner (Reviewer u93d, Reviewer BMVc, Reviewer bjea) and our model effectiveness on non-periodic data (Reviewer aNLj, Reviewer BMVc), we provide explanations as follows. We also carefully consider each comment of Reviewer bjea and realize that there might be some misunderstandings towards our method. We've tried our best to clarify these misunderstandings in the specific rebuttal.

## General Response about frequency channel learner and non-periodic data
**Frequency Channel Learner**

Theoretically, the frequency channel learner is equivalent to global convolutions on the variables for each timestamp. In addition, we have performed visualization experiments to verify the channel-wise learning capability of our frequency channel learner.

+ Theoretically: According to **Theorem 2**, we know that the operations of frequency-domain MLPs can be viewed as global convolutions in the time domain, i.e., Eq. (9) $\mathcal{H}\mathcal{W}+\mathcal{B}=\mathcal{F}(\mathbf{H}\ast W+B)$. As a result, the frequency channel learner containing frequency-domain MLPs can be regarded as global convolutions (global CNNs) over the variable dimension.
+ Empirically, we have investigated the channel-wise learning capability of our frequency channel learner on the METR-LA dataset.

    Specifically, we randomly select 30 detectors and visualize their corresponding adjacency matrix learned by the frequency channel learner via a heatmap. We attach the corresponding heatmap figure in the attached PDF (**see Figure 1 in the attached PDF**). By examining the learned adjacency matrix in conjunction with the actual road map, we can observe that the detectors are very close w.r.t. the physical distance, corresponding to the high values of their correlations with each other in the heatmap. The visualization results demonstrate that the frequency channel learner learns the channel-wise dependencies effectively.

**FreTS on non-periodic data**

Our proposed FreTS is suitable for time-series data that may not be seasonal and may contain some monotonic tendency.

Note that the frequency-domain MLPs compared with time-domain MLPs have two advantages: **global view and energy compaction**.
+ First, the frequency-domain MLPs are equivalent to efficiently performing global convolutions over timestamps (see **Theorem 2**). Accordingly, the frequency temporal learner is able to capture various global temporal information far beyond the periodic information (including periodic patterns, global monotonic tendency, and temporal correlations).
+ In addition, **Theorem 1** implies that, if most of the energy of a time series is concentrated in a small number of frequency components, learning in the frequency spectrum can facilitate preserving clearer/more effective patterns. This characteristic is beneficial for all types of MTS data, regardless of whether the data is periodic or not.

As a result, FreTS is not only suitable for MTS data with periodic patterns, but also for MTS data without periodic patterns.

+ Empirically, our extensive experimental results on different kinds of time series datasets from various applications demonstrate the state-of-the-art performance of our proposed FreTS. The benchmark datasets include Wiki, ECG, and Exchange that do not contain significant seasonal patterns.
+ Moreover, we have provided extensive visualization experiments in **Figures 5, 7, and 8** to investigate the learned weights in the frequency temporal learner. From the results, we observe that the weight coefficients of the real or imaginary part exhibit energy aggregation characteristics (energy compaction). The results on various time series data verify that frequency-domain MLPs can facilitate learning informative features and patterns, which is not limited to time series data with periodic patterns.
+ Furthermore, we have synthesized one multivariate time series dataset (linear: $X=aT+\sigma$ where $a$ is the trend of the variable and $\sigma$ is random noise) and conducted experiments on the dataset. The results under different prediction lengths are visualized in the attached PDF (**see Figures 2, 3, 4, 5, and 6 in the attached PDF**).
From these figures, we can find that FreTS can capture a monotonic tendency.

We hope our response has addressed all concerns. We would greatly appreciate any further constructive comments or discussions.

---

### Decision · Program_Chairs · 2023-09-21

**Decision:**

Accept (poster)

**Comment:**

Dear SAC,

This paper has received the scores of 5, 6, 6, 6, 7, where all reviewers unanimously voted for accepting the paper.

Given the originality and the novelty of the proposed approach of utilizing MLPs in the frequency domain, as well as the relatively high review scores from the reviewers, I recommend to accept the paper as a poster.

Best,

AC